# TEST: Text Prototype Aligned Embedding to Activate LLM's Ability for Time Series

**Chenxi Sun**[1,2,3], **Hongyan Li**[1,2,3,4,*], **Yaliang Li**[5], **Shenda Hong**[6,7,*]

[1]National Key Laboratory of General Artificial Intelligence, Peking University
[2]Key Laboratory of Machine Perception (Ministry of Education), Peking University
[3]School of Intelligence Science and Technology, Peking University
[4]PKU-WUHAN Institute for Artificial Intelligence
[5]Alibaba Group
[6]National Institute of Health Data Science, Peking University
[7]Institute of Medical Technology, Health Science Center of Peking University
`{chenxi_sun,leehy}@pku.edu,cn`
`yaliang.li@alibaba-inc.com, hongshenda@pku.edu.cn`

## Abstract

This work summarizes two ways to accomplish Time-Series (TS) tasks in today's Large Language Model (LLM) context: *LLM-for-TS (model-centric)* designs and trains a fundamental large model, or fine-tunes a pre-trained LLM for TS data; *TS-for-LLM (data-centric)* converts TS into a model-friendly representation to enable the pre-trained LLM to handle TS data. Given the lack of data, limited resources, semantic context requirements, and so on, this work focuses on TS-for-LLM, where we aim to activate LLM's ability for TS data by designing a TS embedding method suitable for LLM. The proposed method is named `TEST`. It first tokenizes TS, builds an encoder to embed TS via instance-wise, feature-wise, and text-prototype-aligned contrast, where the TS embedding space is aligned to LLM's embedding layer space, then creates soft prompts to make LLM more open to that embeddings, and finally implements TS tasks using the frozen LLM. We also demonstrate the feasibility of TS-for-LLM through theory and experiments. Experiments are carried out on TS classification, forecasting, and representation tasks using eight frozen LLMs with various structures and sizes. The results show that the pre-trained LLM with `TEST` strategy can achieve better or comparable performance than today's SOTA TS models and offer benefits for few-shot and generalization. By treating LLM as the pattern machine, `TEST` can endow LLM's ability to process TS data without compromising language ability. We hope that this study will serve as a foundation for future work to support TS+LLM progress.

## 1 Introduction

Implementing Time-Series (TS) tasks, such as medical, industrial, and meteorological, is a research-intensive field Sun et al. (2020). The relevant models evolved from statistical models to RNNs, CNNs, and Transformers. Nowadays, we see a fast growth and remarkable performances of Large-scale pre-trained Language Models (LLM) in NLP and CV fields Zhao et al. (2023). Consequently, it seems natural to inquire whether LLMs can be used for TS tasks. However, according to experiments, most pre-trained LLMs have not made significant progress in relation to abstract TS.

In answer to this requirement, we envision two ways to achieve the paradigm of TS+LLM [1]:

- LLM-for-TS (model-centric, modify LLM). For TS data, design and train a fundamental Large Model from scratch (LM-of-TS), then fine-tune the model accordingly for various downstream tasks. Or, fine-tune the existing pre-trained LLM and convert it from text tasks to TS tasks;

---

[*]Corresponding authors
[1]This categorization focuses on the requirement for changing the model. But from technology, LLM+TS can be achieved by pre-training, fine-tuning, tool-augmented methods, external encoders, and their ensemble.

- TS-for-LLM (data-centric, modify TS). Based on the existing LLMs, furthest freezing them, design some mechanisms to customize TS for them by creating LLM-friendly TS representation.

We acknowledge that the first way, particularly developing and training a model from scratch, is the most essential solution since pre-training is the crucial step of instilling knowledge to the model. And the second way is actually challenging to break beyond the model's original capabilities. However, in this work, we still focus on the second way due to the following three considerations:

*Data perspective.* LLM-for-TS methods, especially when building a foundation model, necessitate large dataset, but TS is professional, the largest dataset is less than 10GB, which is much smaller than that for NLP Zhou et al. (2023); TS-for-LLM methods can use a relatively small dataset as its objective is solely to assist the existing LLM in inferring TS; *Model perspective.* LLM-for-TS methods focus on vertical industries. Because of the major disparities in TS across domains, various large models targeting medical TS, industrial TS, etc. must be built and trained from the start; TS-for-LLM methods need little or even no training. By utilizing plug-in modules, it makes the utilization more general and convenient; *Usage perspective.* LLM-for-TS methods are appropriate for instances involving specialists; TS-for-LLM methods maintain LLM's textual capabilities while providing rich complementing semantics, being easily accessible and user-friendly.

Without changing the existing model, the most natural approach is treating TS as text data. For example, a possible dialogue is: *[Q] Diagnose if a patient has sepsis through the following mean arterial pressure sequence in mm Hg: 88, 95, 78, 65, 52, 30. [A] Yes.* However, TS is often multivariate while text is univariate. For example, excepting mean arterial pressure, dozens of vital signs, and laboratory values, such as heart rate, lactic acid, etc., need to be included when diagnosing sepsis. One intuitive method is to divide a multivariate TS into multiple univariate sequences and input them into LLM one by one. However, this will lead to three drawbacks. First, different prompt sentences, data order, and connection statements will produce different results; Second, a long input sequence likely to make LLM inefficient and hard to remember the previous univariate TS; Third, the crucial aspects of multivariate dependency in TS will be ignored.

To address the above issues and achieve TS-for-LLM, we do not directly input TS into LLM, but instead, we first tokenize TS, then design an encoder to embed them, finally skip the embedding layer to input them into LLM. In this way, the core is to create embeddings that the LLM can understand.

High-quality TS embedding can be employed as the computational phenotype that the deep learning model can understand Hong et al. (2023). To make the embedding understandable by language models. Most multimodal approaches use alignment, for example, aligning text embedding and image embedding through text descriptions of the image Wang et al. (2023). However, TS lacks visual cues and has an annotation bottleneck caused by its complex characteristics. Only a few specific TS, such as ECG, have text descriptions in each segment, where the image-text matching route could be implemented. But in most cases, it's not feasible.

Contrastive Learning (CL) can avoid the annotation bottleneck through designing pretext tasks by utilizing intrinsic information instead of relying on pre-defined prior knowledge. Currently, CL methods for TS data has also advanced Meng et al. (2023b). These methods evaluate the effectiveness of TS embedding through follow-up classification, prediction, or clustering models, such as SVM Franceschi et al. (2019). However, these simple and newly-trained models are considerably different from the complex and pre-trained LLM. The representation vector generated by unconstrained CL is likely to deviate greatly from the LLM's cognitive embedding space.

To address the above issues, we propose an embedding method for `TimE Series` tokens to align the `Text` embedding space of LLM (`TEST`). Based on CL, `TEST` uses text embedding vectors as prototypes to constrain TS' embedding space and highlights feature-wise patterns. We show that `TEST` can activate LLM's ability as pattern machine. The contributions of this work are:

- Summarize two TS+LLM paradigms, LLM-for-TS, TS-for-LLM, with their potential methods;
- Propose `TEST` for TS-for-LLM. `TEST` can produce the similarity-based, instance-wise, feature-wise, and text-prototype-aligned embedding for TS tokens. We prove that prompt tuning is almost equivalent to supervised fine-tuning when TS embedding and word embedding are aligned;
- Experiments on TS classification, forecasting, few-shot, and representation tasks demonstrate that `TEST` can activate LLM's capability to archive TS tasks, where the random and unsatisfactory results produced by original LLMs can be elevated to the baseline.

| Category | Means | Pros | Cons | Work |
|----------|-------|------|------|------|
| LM-of-TS | Training | Specialized, accurate | Not universal, large datasets | Pre-training Ma et al. (2023) Earth transformer Bi et al. (2023) |
| LLM-for-TS | Tuning | End-to-end, accurate | More experiments, lose language ability | GPT4TSZhou et al. (2023) LLM4TSChang et al. (2023) |
| | Tool augmented | Parameter-efficient, less experiments | Need experts, need annotation | PromptCast Xue & Salim (2023) Health Learner Liu et al. (2023) METS Li et al. (2024) Text2ECGChung et al. (2023) |
| TS-for-LLM | External encoder | Parameter-efficient, multiple abilities | Weak robust | TEST |

Table 1: Existing Work about TS+LLM

As the name of TEST implies, it's a forward-looking test that we hope to lay the groundwork for future study. And it does give LLM new capabilities and highlight its qualities as a pattern machine.

## 2 RELATED WORK

### 2.1 TIME SERIES AND LARGE LANGUAGE MODEL

There hasn't been much research done on TS+LLM because this field is still in its infancy. We summarize the existing work in Table 1. LLM-for-TS with changing the model can be achieved through tuning or tool augmented means; TS-for-LLM with changing the data can be achieved through building the external encoder.

LM-of-TS Ma et al. (2023) trains a fundamental and accurate model based on accumulated domain TS data, but it can be difficult to construct a large well-labeled dataset due to data acquisition and annotation costs. By comparison, Supervised Fine-Tuning (SFT) in LLM-for-TS Chang et al. (2023) has a relatively smaller workload than pre-training, but it can make the LLM lose its language capabilities and its advantages over a sophisticated model designed specifically for TS tasks are unclear. Regarding TS as the text sequence and using prompts as the augmented tool Liu et al. (2023) could input numerical TS into LLM directly, but it is inaccurate, requires more experience, and will fail for multivariate TS. The multimodal methods Li et al. (2024) could align the text and TS, but apart from ECG, most TS datasets have no segment annotation.

### 2.2 TIME SEIRES EMBEDDING

TS embedding can provide identities by including typical, associated, and dependant attributes. CL-based methods can get the data representation Chen et al. (2020), employing the instance discrimination pretext task to bring similar pairs closer while pushing dissimilar pairs apart in the embedding space. Some efforts have been made to implement instance-level contrast Woo et al. (2022b); Zheng et al. (2023), temporal-level contrast Meng et al. (2023c); Franceschi et al. (2019), and clustering-level contrast Meng et al. (2023a) on TS data, with promising results. However, the direct contrast cannot bridge TS embedding and the LLM's comprehensible space. In our setting, we prefer to freeze the pre-trained LLM and let the embedding compromise. That is, we use the text token embedding in LLM to limit and guide the TS token embedding.

Inspired by the prototype-level contrast Caron et al. (2020), which goes beyond the independence assumption and exploits latent cluster information present within samples. We can select some text embeddings as basic prototypes to lead the learning. However, in addition to the alignment, we still need to consider issues of prototype selection, differentiation Meng et al. (2023c), uniformity Wang & Isola (2020), stability Huang et al. (2023) and etc.

## 3 METHODS

TEST has two key steps: In Figure 1, build an encoder to embed TS; In Figure 2, create prompts to make the LLM can accept TS embeddings as input.

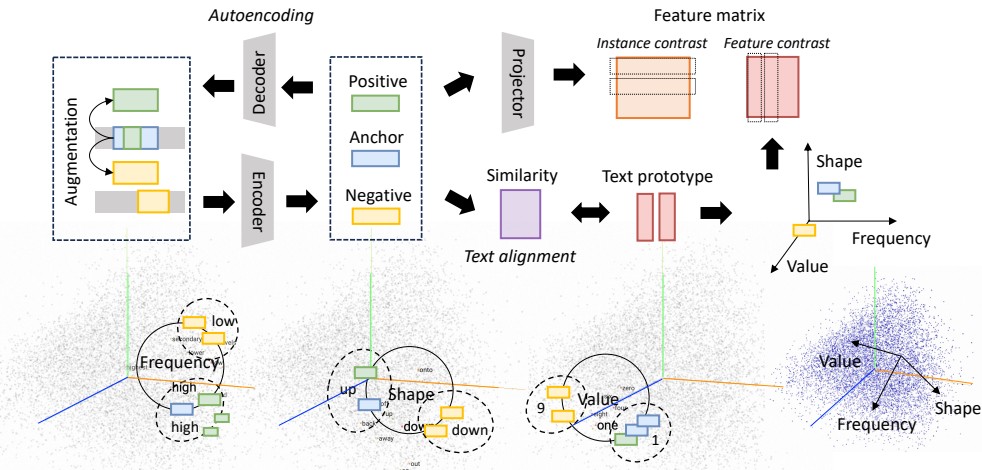

Figure 1: Text-prototype-aligned TS Embedding by Instance-wise and Feature-wise Contrast

## 3.1 TS Token Augmentation and Encoding

**Definition 1 (Token Embedding of Time Series)** *A multivariate time series $x = \{x_t^d\}_{t=1,d=1}^{T,D}$ has $D$ variables and $T$ time points. It can be segmented to a list of $K$ non-overlapping subsequences $s = \{s_k\}_{k=1}^K$ by a segmentation function $f_s : x \to s$, where the length of $s_k = x_{t_i:t_j}$ is arbitrary, $1 \le t_i < t_j \le T$. We call $s$ as the token list of time series $x$. Further, each token can be embedded to a $M$-dimensional representation space by an embedding function $f_e : s_k \in \mathbb{R}^{D \times T} \to e_k \in \mathbb{R}^M$. Finally, the token embedding list of $x$ is $e = \{e_k\}_{k=1}^K = f_e(s) = f_e(f_s(x))$.*

We first tokenize TS into some segmentation/subsequences/tokens/instances through the classical sliding window method in representation learning Yue et al. (2022) $s = f_s(x)$. We define a TS token $s$ as the anchor instance. Its positives $s^+$ are the augmented instances, $s^{weak} \sim \mathcal{T}_{weak}$ (jitter-and-scale strategy, adding random variations to the signal and scale up its magnitude), $s^{strong} \sim \mathcal{T}_{strong}$ (permutation-and-jitter strategy, splitting the sequence into a random number of segments and randomly shuffling them) Eldele et al. (2021). Its negatives $s^-$ are from non-overlapping instances which do not have the same subsequence as $s$.

After getting anchor-positive-negative, we built a neural network as the encoder to embed instance into vector $e = f_e(s)$. We also trained a decoder $f_d$ by using the auto-encoding loss $\mathcal{L}_{ae} = \frac{1}{N} \sum_{i=1}^N \text{sim}(s, f_d(e))$ to ensure the representativeness of the embedding and subsequent verification. Because our primary goal is to retrieve the encoder, this decoder can likewise be unbuilt without harming the future process.

## 3.2 Instance-wise and Feature-wise Contrast

The basic instance-wise CL treats each instance independently and design the instance discrimination pretext task to keep similar instances close and dissimilar instances far away. To prevent embedding space collapse, we treat augmented views of the same instance as the unique positive pair, and all remaining ones within the $B$ size minibatch as negative pairs He et al. (2020). The instance-wise contrastive loss is shown in Equation 1. Where given the instance embedding $e, e^{+/-}$, we construct a projection head $f_p$, which is a one-layer MLP to obtain $f_p(e)$. $\sigma(e, e^{+/-})$ is used to calculate the similarity between two projected vectors through a similarity function $\text{sim}$ like cosine similarity with the instance-level temperature parameter $\tau$.

$$\mathcal{L}_{ins} = -\log \frac{\exp(\sigma(e, e^+))}{\exp(\sigma(e, e^+)) + \sum_{i=1}^B \exp(\sigma(e, e_i^-))}$$
$$\sigma(e, e^{+/-}) = \frac{\text{sim}(f_p(e), f_p(e^{+/-}))}{\tau} \tag{1}$$

We also propose a feature-wise contrast method to break the independence between instances. As shown in Figure 1, after embedding, a feature matrix $\mathbb{R}^{B \times M}$ is formed by the representation vectors of instances in a minibatch. Where each row is an embedding of a instance, thus rows could be regarded as soft labels of instances which are used in Equation 1. In addition to rows, columns of feature matrix also have semantic information. Li et al. (2021) proposed that the columns could be further regarded as cluster representations. However such cluster-wise methods require prior knowledge to pre-specify the number of clusters, which is non-trivial for the unlabeled TS data in this work. Thus, we propose to regard the columns as the soft labels of features and perform discrimination between groups of similar features.

For an anchor feature matrix m, where m is the $B$-th row copy of the vector $e$, we obtain a positive feature matrix $\mathrm{m}^+$ and a negative feature matrix $\mathrm{m}^-$, where $\mathrm{m}^{+/-} = [e_i]_{i=1}^B \in \mathbb{R}^{B \times M}$. We mark the columns in the matrix as $m \in \mathrm{m}^\mathrm{T}$. As expressed by the item before the right arrow in the Equation 2, the feature-wise contrast mainly align and differentiate the same feature column among the positive and negative. However, this may cause the representation space to shrink within a small area. We find that ensuring differences between features can better address this issue. That is, we suggest the contrast between different feature columns as shown in the item after the right arrow.

$$\mathcal{L}_{fea} = -\sum_{i=1}^{M}(\underbrace{\sigma(m_i, m_i^+)}_{\text{Alignment}} - \underbrace{\sigma(m_i, m_i^-)}_{\text{Difference}}) \Rightarrow -\sum_{i=1}^{M}\underbrace{\log \frac{\exp(\sigma(m_i, m_i^+))}{\sum_{j=1}^{M}[\exp(\sigma(m_i, m_j^+)) + \exp(\sigma(m_i, m_j^-))]}}_{\text{Feature category uniformity}} \quad (2)$$

More importantly, the injection of feature column differences can also greatly assist in the subsequent implementation of text-prototype-aligned contrast. Because that contrast will apply the selected text token embedding to the feature columns, like coordinate axes.

### 3.3 TEXT-PROTOTYPE-ALIGNED CONTRAST

The pre-trained LLM has its own token embedding, e.g., small, medium, and big GPT-2 embed text tokens from word dictionaries into representation spaces with 768, 1024, and 1280 dimensions. Naively, we can align the token embedding of TS and text using the similarity estimation. Although TS tokens lack text annotation, we can place their embedding near typical text descriptions of TS, such as value, shape, and frequency. In this fashion, it is intuitively expected that various TS tokens can represent various descriptive terms such as small, big, up, down, stable, fluctuating, and so on. Naturally, the example above is based on the closest neighbor principle because the embedding space of a text token is discrete, akin to a vector table, but that of our TS token is continuous.

However, of course, the actual outcomes will not match what we expect because we are not providing the supervised label or ground truth. For example, the embedding of a subsequence with an upward trend may be very close to that of a decline word, or even that does not describe the trend. But it is irrelevant whether semantics can be understood by us. As usual, the fact is that humans cannot comprehend the model's perceptual mode.

Recently, researchers proved that LLMs are pattern machines Mirchandani et al. (2023). Thus, in this work, we achieve "TS → pattern → text" to activate LLM's ability for TS tasks. The choice of text prototype can be relaxed, not necessarily the description related to TS.

In this work, we choose $P$ representative text embedding $tp$ as pivots/prototypes, and map TS embedding to them. In high dimensional space, almost all vectors are pairwise orthogonal Hopcroft & Kannan (2013), thus the number of prototypes rather than the type does matter, and their differences can be reflected in a single dimension/feature. Thus, the modeling function of the text prototype $tp$ is realized by feature-wise contrast. As expressed by Equation 3, the alignment term guarantees that the two space ranges are roughly the same through the similarity constraint, the contrast term uses $tp$ as the coordinate axis to map the TS embedding, making the representation values in text coordinate axes of similar instance similar. The feature matrix is no longer obtained through the projector but through the prototype mapping $e \cdot tp \to \mathrm{m}$.

$$\mathcal{L}_{text} = -\sum_{i=1}^{P}[\underbrace{\text{sim}(tp_i, e)}_{\text{Text alignment}} - \underbrace{\mathcal{L}_{fea}(e \cdot tp, e^+ \cdot tp, e^- \cdot tp)}_{\text{Text contrast}}] \quad (3)$$

### 3.4 LEARNABLE PROMPT EMBEDDING

Even TS has been described using an embedded representation that the LLM can understand, LLM still has to be instructed on how to do subsequent TS tasks.

Prompt engineering like template and chain-of-thought is intuitive. Their contexts are coherent in human semantics, but a TS embedding list has no human semantics, it is more about a pattern sequence. Thus, to create a more consistent prompt pattern, we train a soft prompt by p-tuning Lester et al. (2021) make LLM be easier to understand the input. These soft prompts are task-specific embedding, learning through the loss from LLM's output and task ground truth in Equation 4.

$$\mathcal{L}_{promp} = L_{reg/cls}(\text{concat}(pe, e)) \quad (4)$$

GPT4TS Zhou et al. (2023)has proved the feasibility that SFT can make LLM apply to TS. Based on this, we demonstrate the feasibility of TEST by proving the equivalence between soft prompt and SFT.

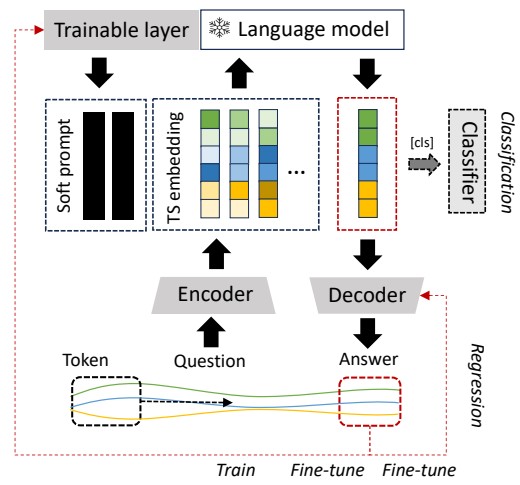

Figure 2: Framework of LLM for TS Tasks

Consider a conditional generation task where the input $x$ is a context and the output $y$ is a sequence of tokens. Assume an autoregression LLM $p_\phi(y|x)$ with parameter $\phi$, $z = [x; y]$. The inference of a pre-trained LLM is computing $h_i$ as a function of $z_i$ and the past activations in its left context, $Y = \mathcal{LM}_\phi(z_i, h_i)$. The past $h_i$ in the soft prompt turning with prompt $pe_\theta$ is $h_i = \begin{cases} pe_\theta[i, :], & \text{if } i \in pe_{\text{idx}} \\ \mathcal{LM}_\phi(z_i, h_i), & \text{otherwise} \end{cases}$. The SFT from LLM to TS-LLM is Equation 5. Its transformation shows that the soft prompt tuning is approximately equivalent to SFT.

$$
\begin{aligned}
\max_\phi p_\phi(y'|x) = \max_\phi \sum_{i \in Y_{\text{idx}}} \log p_\phi(z'_i|h_{<i}) &= \sum_{i \in Y_{\text{idx}}} \log p_{\phi+\Delta}(z_i + \delta z_i|h_{<i}) \\
&\approx \sum_{i \in Y_{\text{idx}}} \log p_\phi(z_i|h_{<i}) \cdot \sum_{i \in pe_{\text{idx}}} \log p_\Delta(\delta z_i|h_{<i}) \\
&= \underbrace{\sum_{i \in Y_{\text{idx}}} \log p_\phi(z_i| \underbrace{f_e(s)}_{\text{Text-TS alignment}})}_{\text{Frozen LLM}} \cdot \underbrace{\sum_{i \in pe_{\text{idx}}} \log p_\Delta(\delta z_i|h_{<i})}_{\text{Prompt } pe_\theta}
\end{aligned}
\quad (5)
$$

Equation 5 also suggests that the projection space of TS tokens should preferably cover the complete set of text embedding space. Thus, we utilize clustering to find $P$ representative text prototypes. The process of using LLM to infer TS is shown in Figure 2. In this framework, the text data is input into the embedding layer of LLM, while the prompts and TS embeddings skip this layer.

## 4 EXPERIMENTS

The core of TEST is to train an encoder $f_e$ and a soft prompt $pe$ as described in Algorithm 1. The encoder must can extract relevant information from TS, needs to be time- and memory-efficient, and has to allow variable-length inputs. Thus, we build a causal TCN with 10 layers of convolution blocks. Each convolution block is a sequence of GELU, DilatedConv, BatchNorm, GELU, DilatedConv, with skip connections across each block. The DilatedConvs have dilation of $2i$ in each layer $i$ of convolution block. A final convolution block is used to map the hidden channels to the output channel whose size is the same as the LLM's embedding size.

---

**Algorithm 1** Training `TEST`

---

| 1: **for** e in epochs **do** | 9: **for** e in epochs **do** |
|---|---|
| 2:     // UPDATE ENCODER | 10:     // UPDATE PROMPT |
| 3:     $\theta_{f_e} = \theta_{f_e} - \eta \nabla_{\theta_{f_e}} (\mathcal{L}_{ins} + \mathcal{L}_{text})$ | 11:     $pe = pe - \eta \nabla_{\theta_{pe}} \mathcal{L}_{promp}$ |
| 4:     // UPDATE DECODER (OPTIMAL) | 12:     // FINE TUNE DECODER (OPTIMAL) |
| 5:     $\theta_{f_d} = \theta_{f_d} - \eta \nabla_{\theta_{f_d}} \mathcal{L}_{ae}$ | 13:     $\theta_{f_d} = \theta_{f_d} - \eta' \nabla_{\theta_{f_d}} \mathcal{L}_{reg}$ |
| 6:     // UPDATE PROJECTOR | 14:     // UPDATE CLASSIFIER (OPTIMAL) |
| 7:     $\theta_{f_p} = \theta_{f_p} - \eta \nabla_{\theta_{f_p}} \mathcal{L}_{ins}$ | 15:     $\theta_{f_c} = \theta_{f_c} - \eta \nabla_{\theta_{f_c}} \mathcal{L}_{cls}$ |
| 8: **end for** | 16: **end for** |

---

The used LLMs are as listed in Table 2. Each encoder and soft prompt of LLM are trained using the Adam optimizer on 20 NVIDIA Tesla V100-SXM2 GPU with CUDA 11.3.

| Model | Size | Embed. dimension |
|---|---|---|
| Bert Devlin et al. (2018) | 110M, 335M | 748, 1024 |
| GPT2 Radford et al. (2019) | 117M, 345M, 774M | 768, 1024, 1280 |
| ChatGLM Du et al. (2022) | 6B | 4096 |
| LLaMa2 Touvron et al. (2023) | 7B, 13B | 4096 |

Table 2: The Used Language Model

We compare our method to 5 kinds of methods including 12 baselines: 1) LLM-QA methods Xue & Salim (2023); Liu et al. (2023) with the classification template *Classify the given [domain] sequence as either [class label] or [class label]: [numerical sequence]. [A]* and the forecasting template *[Q] Forecast the next value of the given [domain] sequence: [numerical sequence]. [A]*; 2) SFT LLM-for-TS method GPT4TS Zhou et al. (2023); 3) classical TS models DWT, DWTD Bagnall et al. (2018), 1NNED, and TCN Tan et al. (2021); 4) SOTA TS models Informer Zhou et al. (2021), DLinear Zeng et al. (2023), and TimesNet Wu et al. (2023); 5) SOTA CL-based TS models Tloss Franceschi et al. (2019), TS2Vec Yue et al. (2022), and CoST Woo et al. (2022a).

The overall results are shown in Figure 3 (The appendix has more compared classical SOTA models and detailed results about long-term, short-term, few-shot, and zero-shot forecasting, multivariate time series classification, and representation tasks.). Overall, after using `TEST`, when the size of LLM reaches about 300M, their accuracy comparable to SOTA model.

## 4.1 CLASSIFICATION

We present accuracy scores for all 128 kinds of univariate TS datasets in UCR archive Dau et al. (2019) and all 30 kinds of multivariate TS datasets in UEA archive Bagnall et al. (2018).

**Accuracy.** In Figure 3 (a-b), `TEST` makes the classification accuracy of LLM increase significantly. LLM's original classification performances are demonstrated through two QA results. It almost guesses the classification labels at random, especially for multivariate TS. After using `TEST`, GPT2-774M, which has the median accuracy among all models, can improve accuracy by at least 18% for univariate TS and 25% for multivariate TS. `TEST` makes most LLMs comparable to, if not better than, the existing models. When the size reaches about 300M, the accuracy can exceed TS baselines; When the size reaches about 700M, the accuracy can exceed SOTA TS transformers.

**Ablation.** In Figure 3 (c-d), different text prototypes will lead to different results. We set 3 groups of text prototypes: embeddings of *value*, *shape*, *frequency*, and embeddings of 3 or 10 cluster centers. Choosing a prototype group that more accurately represents LLM's entire text embedding space can improve the performance. This is also suggested by Equation 5. Different prompt types, initialization, and length will lead to different results. We compare the soft prompt with the hard prompt of *Classify the given [domain] sequence as either [class label] or [class label]: [TS embedding]*. The accuracy differs by at least 10%. We set random initialization from uniform distribution and task description initialization from *Classify the given sequence*. The latter makes the training converge faster. When the model reaches 1B, a prompt length of 10 can achieve excellent results.

## 4.2 FORECASTING

We present short-forecasting MSE scores for all 19 kinds of varied time series datasets in TSER archive Tan et al. (2021), and long-forecasting MSE scores for 8 popular real-world benchmark datasets including weather, traffic, electricity, ILI, and ETT from Wu et al. (2023).

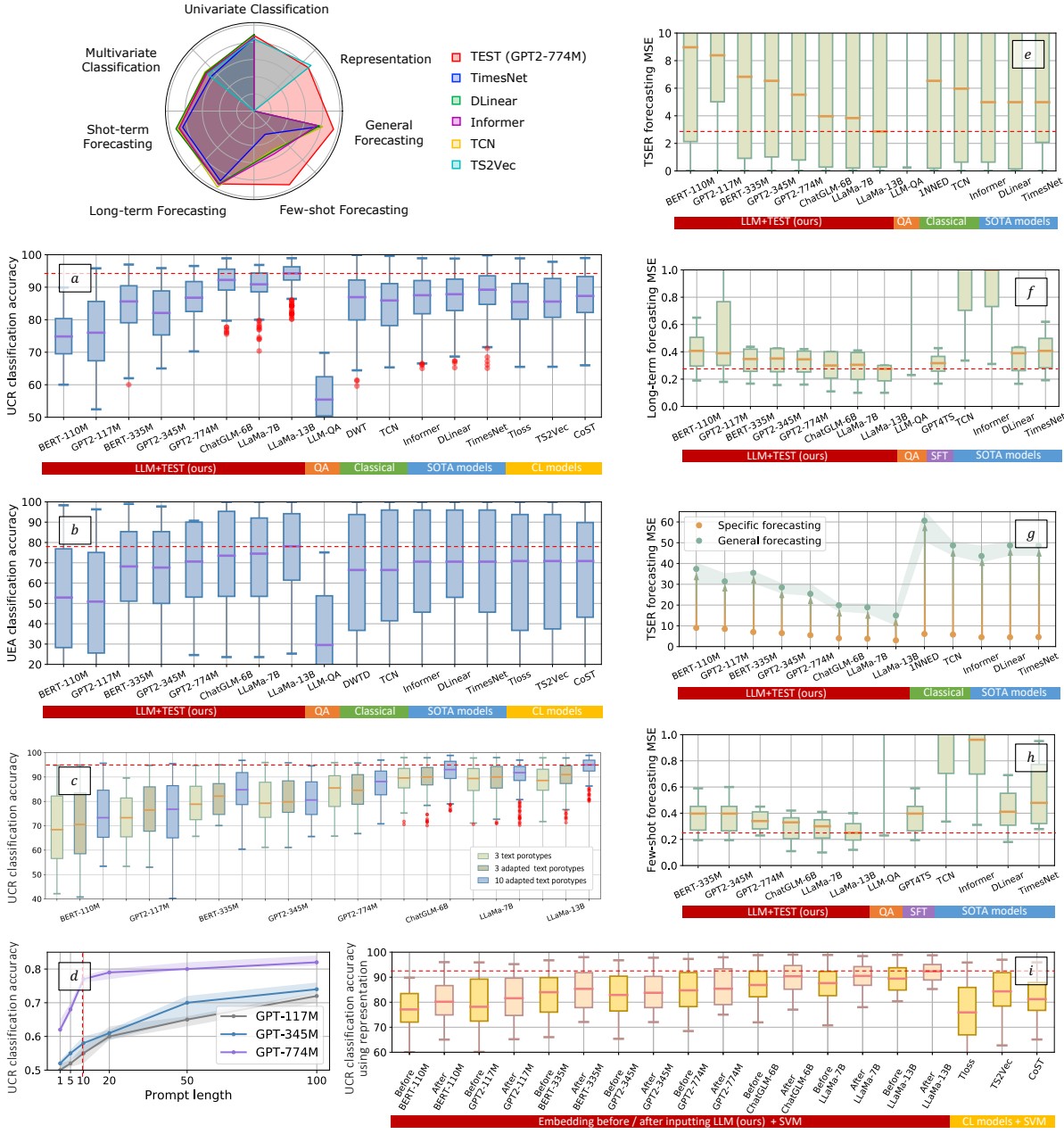

Figure 3: Experiment Results. (a-d) shows the classification results; (e-h) shows the forecasting results; (i) shows the representation results. The red dashed line represents the best result.

**Accuracy.** In Figure 3 (e-f), TEST makes the forecasting accuracy of LLM increase significantly and comparable to SOTA models. When the size reaches about 300M, the accuracy can exceed SOTA TS transformers.

**Generalization.** We fuse 19 datasets into 1 dataset and test the method on this fused dataset. As shown in Figure 3 (g), compared with baselines, LLM-based models have better generality.

**Few-shot.** LLM has demonstrated remarkable performance in few-shot learning. Based on the settings in Zhou et al. (2023), we present few-shot forecasting for 10% time steps in training datasets. As shown in Figure 3 (h), TEST achieves the best performance and demonstrates a relative average MSE reduction of 23.5%.

### 4.3 Representation

**Representation learning.** Learning universal representations for TS is a fundamental but challenging problem. Both `TEST`'s first step (creating TS embedding) and second step (LLM's output) can achieve this task. Based on the classical representation learning task, we evaluated the effectiveness of `TEST` representation using SVM classifier on UCR dataset. Note that using a simple classifier can better reflect the presentation effect. In Figure 3 (i), the embedding in `TEST`'s first step is comparable to SOTA representation methods, and the embedding in `TEST`'s second step can outperform them. This indicates that after using LLM, the representation of TS becomes more discriminative.

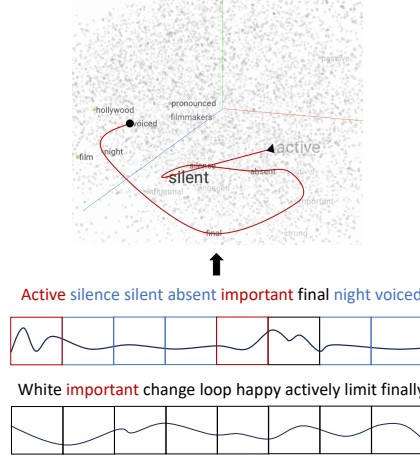

Figure 4: Matching TS Embedding to Words

**Case.** We use nearest neighbor method to find the text that a TS token matches to in the word embedding space of frozen LLM. In Figure 4, the majority of the identified words are sentiment-related adjectives and nouns. We speculate that by prompting, the model will treat TS classification task as an sentiment classification task. Thus, introducing prompt is like introducing a shortcut for LLM. Besides, the matched words are like a kind of textual Shapelet for TS segmentation, representing TS through a series of patterns. Instead of regarding TS as a sequence of numbers, we suggest using words to identify patterns in TS as LLMs without SFT are not good for math when performing digital tasks, but they are good at extracting knowledge as a pattern machine. The semantics of the patterns be perplexing to us, but it makes sense to LLM.

## 5 Discussion and Conclusion

This paper proposes an instance-wise, feature-wise, and text-prototype-aligned TS embedding method to achieve TS-for-LLM. It can activate LLM's ability for TS tasks while maintaining its original language ability. Experiments on classification, forecasting, and representation tasks show that using `TEST`, LLM can archive comparable performance to SOTA methods.

*TS-for-LLM can enrich LLM's capabilities.* SFT LLM may be more effective than TS-for-LLM, yet its superiority over customized TS models remains unclear; Training customized models may be more accurate in TS tasks, yet TS-for-LLM offers all notable benefits of LLM additionally.

*TS-for-LLM can explore LLM's mechanism as a pattern machine.* The essence of TS-for-LLM is: TS ↔ TS embeddings ↔ patterns ↔ text/word embedding ↔ text. Although `TEST` gives the impression of a forcibly aligning operations between TS and text, it dose convert TS into an understandable pattern sequence for LLMs, that clearly demonstrates that the essence of LLM is pattern recognition. In fact, TS is objective data, whereas images, text, and speech are subjective data that can be perceived by human senses. `TEST` aligns objective TS data and subjective text data at the machine level, but how to align them at the human perception level requires future research.

Meanwhile, in addition to text prototypes and prompts, LLM size and type also affect the results. The impact of model type is intuitive, it is related to downstream tasks, where the bidirectional structure is beneficial for classification, and the generated structure is beneficial for forecasting. The impact of model size, where a larger model produces more accurate results, can be attributed to various reasons. Aside from the impact of additional parameters, we believe that the datasets used in the pre-training process are also important, with the size, diversity, and corpus type all having an impact. We conjecture that more training data will provide the model with more opportunities to learn temporal patterns. As a result, we intend to conduct more experiments to investigate deeper correlations between corpora and TS data Chen et al. (2023).

### Acknowledgments

This work is supported by National Natural Science Foundation of China (No.62172018, No.62102008) and Wuhan East Lake High-Tech Development Zone National Comprehensive Experimental Base for Governance of Intelligent Society. We extend our heartfelt gratitude to Eamonn's lab and THUML for generously providing the UCR datasets and TSlib datasets utilized in this study.

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

# APPENDIX

## 1 RELATED WORK

Our work mainly involves two research fields: Universal Representation Learning (URL) for time series based on Contrastive Learning (CL) and Large Language Model (LLM) + Time Series (TS).

### 1.1 CL-BASED URL FOR TS

Unsupervised URL approaches aim to learn discriminative feature representations from unlabeled data, without the requirement of annotating every sample. Enabling URL is extremely crucial for time series data, due to its unique annotation bottleneck caused by its complex characteristics and lack of visual cues compared with other data modalities.

Contrastive methods learn meaningful representations from time series by optimizing self-discrimination tasks. Instead of directly modeling the complex raw data, they employ pretext tasks that leverage the underlying similarity between samples, which eliminates the need for reconstructing the complete input and allows for the discovery of contextualized underlying factors of variations. Contrastive methods typically generate augmented views of the raw data through various transformations and then learn representations by contrasting positive samples against negative samples. The existing CL-based URL for TS are listed in Table S2.

Instance-level contrastive models treat individual samples independently for the purpose of instance discrimination. They utilize data augmentations to transform original inputs into a new embedding space. Within this space, augmentations derived from the same sample are considered as positive pairs, while those from different samples are treated as negative pairs. During training, these models are optimized by maximizing the similarity between representations of positive pairs, while simultaneously minimizing the similarity between representations of negative pairs.

Prototype-level contrastive models break the independence between samples and explore to exploit the implicit semantics shared by samples in the same cluster. They can address the limitation that instance-level contrastive learning models tend to treat semantically similar samples as negatives.

Temporal-level contrastive models instead focus on capturing scale- invariant representations at each individual timestamp. By cosidering both instance-level and temporal-level representation learning strategies, researchers aim to enhance the capability of contrastive learning methods in capturing the complexities inherent in time series data.

| Category | Pros | Cons | Methods |
|---|---|---|---|
| Reconstruction-based | Disregard insignificant data that may contain noise | Collapse of embedding space; Unable to measure feature relations | TimeNetWu et al. (2023) SimMTM Dong et al. (2023) |
| Adversarial | Eliminate the need for expensive manual labeling | Difficulty in model convergence; Unable to measure feature relations | TimeGAN Yoon et al. (2019) TS-GAN Brophy et al. (2023) |
| Predicative | Self-supervised | Affected by noise | TST Zerveas et al. (2021) TS-TCCEldele et al. (2021a) |
| Contrastive | Self-supervised | Different datasets require different data augmentation methods and similarity evaluations | Table S2 |

Table S1: Representation Learning Methods of Time Series Methods

| Type | Methods | | | |
|---|---|---|---|---|
| Instance-level | SimCLR Chen et al. (2020) | TimeCLR Yang et al. (2022) | MoCo He et al. (2020) | BYOL Grill et al. (2020) |
| | CPC van den Oord et al. (2018) | SimSiam Zheng et al. (2023) | MCL Wickstrøm et al. (2022) | |
| Prototype-level | SwAV Caron et al. (2020) | PCL Li et al. (2021b) | CCL Sharma et al. (2020) | SCCL Zhang et al. (2021) |
| | CC Li et al. (2021c) | SLIC Khorasgani et al. (2022) | MHCCL Meng et al. (2022) | |
| Temporal-level | TS2Vec Yue et al. (2022) | TS-TCC Eldele et al. (2021b) | TNC Tonekaboni et al. (2021) | TCL |
| | T-Loss Franceschi et al. (2019b) | BTSF Yang & Hong (2022) | CoST Woo et al. (2022a) | |

Table S2: Contrastive Learning based Universal Representation Methods for Time Series

| Means | Pros | Cons | Work |
|---|---|---|---|
| Training | Specialized, accurate | Not universal, large datasets | Pre-training Ma et al. (2023) Earth transformer Bi et al. (2023) TS Transformers Wu et al. (2023) |
| Tuning | End-to-end, accurate | More experiments, lose language ability | GPT4TSZhou et al. (2023) LLM4TSChang et al. (2023) LLMTime Gruver et al. (2023) Time-LLM Jin et al. (2023) |
| Tool Augmented | Parameter-efficient, less experiments | Need experts, need annotation | PromptCast Xue & Salim (2023) Health Learner Liu et al. (2023) METS Li et al. (2024) Text2ECGChung et al. (2023) |
| External Encoder | Parameter-efficient, multiple abilities | Weak robust | `TEST` |

Table S3: Existing Work about TS+LLM

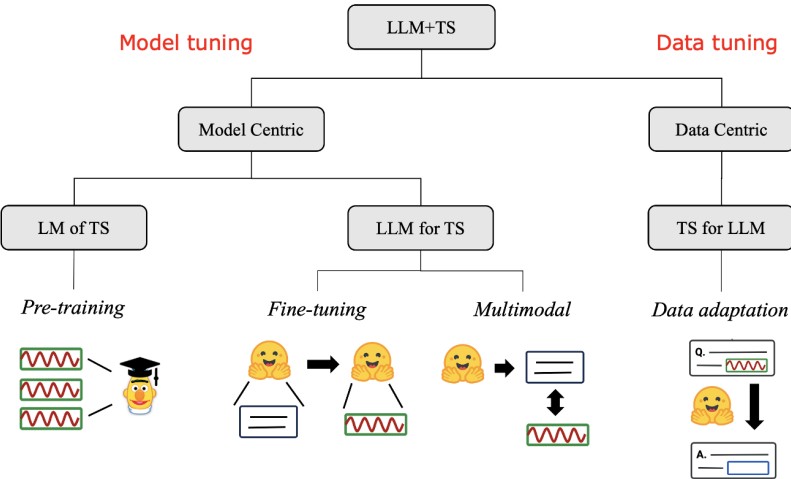

Figure S1: Technical Route of LLM+TS

## 1.2 LLM+TS

Large models, specifically referred to as large language models (LLMs) and pre-trained foundation models (PFMs), have witnessed remarkable success across a multitude of tasks and domains, such as natural language processing (NLP), computer vision (CV). Given the remarkable achievements of large models in these diverse fields, an intriguing question emerges: can large models be effectively employed to analyze TS data?

TS data has long been studied and proven to be indispensable in a myriad of real-world applications, encompassing fields such as geoscience, transportation, energy, healthcare, environment, and finance. While large models have made significant progress in various fields, the arena of time series analysis has followed a more gradual path. Traditional analytical methods have predominantly relied on statistical models. The advent of deep learning has galvanized the research community to

explore more potent data-driven models, typically built on the basis of Recurrent Neural Networks (RNNs), Convolutional Neural Networks (CNNs), and Transformers. Nonetheless, the majority of these models remain relatively small in scale and are tailored for specific tasks, thereby lacking the capacity to acquire comprehensive semantic and knowledge representations from large-scale data for multi-task reasoning.

There hasn't been much research done on TS+LLM because this field is still in its infancy. We summarize the existing work in Table S3. Different from the main text, we category work here through technical means.

## 2 EXPERIMENTS

https://github.com/SCXsunchenxi/TEST

### 2.1 MODEL

#### 2.1.1 ENCODER

The core of TEST is to train an encoder and a soft prompt. The encoder must can extract relevant information from TS, needs to be time- and memory-efficient, and has to allow variable-length inputs. Thus, as shown in Figure S2, we build a causal TCN with 10 layers of convolution blocks. Each convolution block is a sequence of GELU, DilatedConv, BatchNorm, GELU, DilatedConv, with skip connections across each block. The DilatedConvs have dilation of $2i$ in each layer $i$ of convolution block. A final convolution block is used to map the hidden channels to the output channel whose size is the same as the LLM's embedding size.

The detailed architecture is: Number of channels in the intermediary layers of the causal network is $40$; Number of layers (depth of the causal network) is $10$; Kernel size of all convolutions is $3$; Negative slope of the leaky ReLU activation is $0.01$; Number of output channels of the causal network (before max pooling) is $640$; Dimension of the representations is the same as the LLM's embedding size (e.g. 1024 for gpt2).

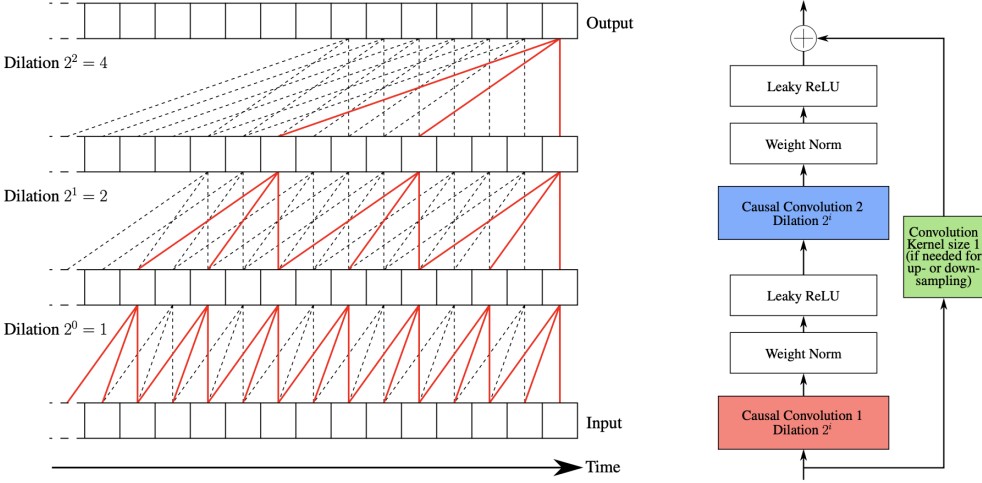

Figure S2: Illustration of Three Stacked Dilated Causal Convolutions and Composition of the i-th Layer of The Chosen Architecture

We train our models with the following parameters for time series classification. Note that no hyperparameter optimization was performed on the encoder hyperparameters: Optimizer is Adam with learning rate $\alpha = 0.001$ and decay rates $\beta = (0.9, 0.999)$; Number of negative samples is $K \in \{1, 2, 5, 10\}$ for for univariate time series, $K \in \{5, 10, 20\}$ for multivariate ones; Batch size is $10$; Number of optimizations steps is 2000for $K \leq 10$ (i.e., 20 epochs for a dataset of size 1000), 1500 otherwise.

### 2.1.2 LLM

The used LLMs are as listed in Table S4. Each encoder and soft prompt of LLM are trained using the Adam optimizer on 20 NVIDIA Tesla V100-SXM2 GPU with CUDA 11.3.

| Model | Size | Embed. dimension |
|---|---|---|
| Bert Devlin et al. (2018) | 110M, 335M | 748, 1024 |
| GPT2 Radford et al. (2019) | 117M, 345M, 774M | 768, 1024, 1280 |
| ChatGLM Du et al. (2022) | 6B | 4096 |
| LLaMa2 Touvron et al. (2023) | 7B, 13B | 4096 |

Table S4: The Used Language Model

## 2.2 FORECASTING TASKS

All the deep learning networks are implemented in PyTorch and trained on NVIDIA V100 32GB GPUs. We use mean square error (MSE) and mean absolute error (MAE) as metrics. For zero-shot learning, mean absolute percentage error (MAPE) is used for TOURISM; symmetric MAPE (sMAPE) is used for M3 and M4; normalized deviation (ND) is used for ELECTR. All experiments are repeated 3 times and the mean of the metrics is used in the final results.

### 2.2.1 DATASET DETAILS

The details of long-term forecasting and few-shot forecasting datasets are: ETT datasets Zhou et al. (2021) contain electricity load of various resolutions (ETTh & ETTm) from two electricity stations; Weather datasetWetterstation (2017) contains 21 meteorological indicators of Germany within 1 year; Illness datasetCDC (2021) contains the influenza-like illness patients in the United States. ILI is not used for few-shot learning for the limited quantity that is hard to follow the definition of few-shot; Electricity dataset SJ & B (2017) contains the electricity consumption; Traffic dataset PeMS (2021) contains the occupation rate of freeway system across the State of California. Table S5 summarizes details of feature statistics.

| Dataset | Length | Dimension | Frequency |
|---|---|---|---|
| ETTh | 17420 | 7 | 1 hour |
| ETTm | 69680 | 7 | 15 min |
| Weather | 52696 | 22 | 10 min |
| ILI | 966 | 7 | 7 days |
| Electricity | 26304 | 321 | 1 hour |
| Traffic | 17544 | 862 | 1 hour |

Table S5: Long-term Forecasting and Few-shot Forecasting Dataset Details

| | Dataset | | Mapping | |
|---|---|---|---|---|
| | Length | Horizon | M4 | M3 |
| M3 Yearly | 645 | 6 | Yearly | - |
| M3 Quarterly | 756 | 8 | Quarterly | - |
| M3 Monthly | 1428 | 18 | Monthly | - |
| M3 Others | 174 | 8 | Monthly | - |
| M4 Yearly | 23000 | 18 | - | Yearly |
| M4 Quarterly | 6 | 24000 | - | Quarterly |
| M4 Monthly | 8 | 48000 | - | Monthly |
| M4 Weekly | 359 | 13 | - | Monthly |
| M4 Daily | 4227 | 14 | - | Monthly |
| M4 Hourly | 414 | 48 | - | Monthly |
| TOURISM Yearly | 518 | 4 | Yearly | Yearly |
| TOURISM Quarterly | 427 | 8 | Quarterly | Quarterly |
| TOURISM Monthly | 366 | 24 | Monthly | Monthly |
| ELECTR | 1311 | 168 | Hourly | Monthly |

Table S6: Zero-term Forecasting Datasets and Mapping Details of Zero-shot Learning

The details of zero-shot forecasting datasets are: M4 is a large and diverse dataset that contains time series of various frequencies and fields, including business, financial and economic forecasting; M3 is smaller than M4, but also contains time series from diverse domains and frequencies; TOURISM is the dataset of tourism activities with different frequencies and contains a much higher fraction of erratic series compared with M4; ELECTR represents the electricity usage monitoring of 370 customers over three years. Table S6 summarizes details of the datasets and zero-shot mapping between source and target.

### 2.2.2 BASELINE DETAILS

For long-shot forecasting, we refer to the SOTA methods reported in Wu et al. (2023): TimesNet Wu et al. (2023), ETSformer Woo et al. (2022b), DLinear Zeng et al. (2023), FEDformer Zhou et al. (2022), Informer Zhou et al. (2021), and LLM for TS method GPT4TS Zhou et al. (2023).

For few-shot forecasting, we refor to the SOTA methods reported in Zhou et al. (2023): DLinear Zeng et al. (2023), PatchTST Nie et al. (2023), TimesNet Wu et al. (2023), FEDformer Zhou et al. (2022), Autoformer Wu et al. (2021), Stationary Liu et al. (2022), ETSformer Woo et al. (2022b), Informer Zhou et al. (2021), Reformer Kitaev et al. (2020)

For zero-shot forecasting, we refor to the SOTA methods reported in Zhou et al. (2023): N-BEATS Oreshkin et al. (2020), DLinear Zeng et al. (2023), PatchTST Nie et al. (2023), TimesNet Wu et al. (2023), FEDformer Zhou et al. (2022), Autoformer Wu et al. (2021), Stationary Liu et al. (2022), ETSformer Woo et al. (2022b), Informer Zhou et al. (2021), Reformer Kitaev et al. (2020)

| Methods | | TEST | GPT4TS | TimesNet | ETSformer | DLinear | FEDformer | Informer | TCN | LSTM |
|---|---|---|---|---|---|---|---|---|---|---|
| ETTm1 | 96 | 0.293 0.346 | 0.292 0.346 | 0.325 0.398 | 0.338 0.375 | 0.345 0.372 | 0.375 0.398 | 0.672 0.571 | 0.863 0.664 | 0.863 0.664 |
| | 192 | 0.332 0.369 | 0.332 0.372 | 0.324 0.387 | 0.408 0.410 | 0.380 0.389 | 0.426 0.441 | 0.795 0.669 | 0.837 0.700 | 1.113 0.776 |
| | 336 | 0.368 0.392 | 0.366 0.394 | 0.360 0.411 | 0.435 0.428 | 0.413 0.413 | 0.445 0.459 | 1.212 0.871 | 1.124 0.832 | 1.267 0.832 |
| | 720 | 0.418 0.420 | 0.417 0.421 | 0.428 0.450 | 0.499 0.462 | 0.474 0.453 | 0.543 0.490 | 1.166 0.823 | 1.153 0.820 | 1.324 0.858 |
| | Avg | 0.353 **0.382** | 0.352 0.383 | **0.350** 0.406 | 0.429 0.425 | 0.403 0.407 | 0.448 0.452 | 0.961 0.734 | 0.929 0.725 | 1.142 0.782 |
| ETTh1 | 96 | 0.372 0.400 | 0.376 0.397 | 0.384 0.402 | 0.494 0.479 | 0.386 0.400 | 0.376 0.419 | 0.865 0.713 | 0.878 0.740 | 1.044 0.773 |
| | 192 | 0.414 0.422 | 0.416 0.418 | 0.436 0.429 | 0.538 0.504 | 0.437 0.432 | 0.420 0.448 | 1.008 0.792 | 1.037 0.824 | 1.217 0.832 |
| | 336 | 0.422 0.437 | 0.442 0.433 | 0.491 0.469 | 0.574 0.521 | 0.481 0.459 | 0.459 0.465 | 1.107 0.809 | 1.238 0.932 | 1.259 0.841 |
| | 720 | 0.447 0.467 | 0.477 0.456 | 0.521 0.500 | 0.562 0.535 | 0.519 0.516 | 0.506 0.507 | 1.181 0.865 | 1.135 0.852 | 1.271 0.838 |
| | Avg | **0.414** 0.431 | 0.427 **0.426** | 0.458 0.450 | 0.542 0.510 | 0.456 0.452 | 0.440 0.460 | 1.040 0.795 | 1.072 0.837 | 1.198 0.821 |
| ETTh2 | 96 | 0.275 0.338 | 0.285 0.342 | 0.340 0.374 | 0.340 0.391 | 0.333 0.387 | 0.358 0.397 | 3.755 1.525 | 2.116 1.197 | 2.522 1.278 |
| | 192 | 0.340 0.379 | 0.354 0.389 | 0.402 0.414 | 0.430 0.439 | 0.477 0.476 | 0.429 0.439 | 5.602 1.931 | 4.315 1.635 | 3.312 1.384 |
| | 336 | 0.329 0.381 | 0.373 0.407 | 0.452 0.452 | 0.485 0.559 | 0.594 0.541 | 0.496 0.487 | 4.721 1.835 | 1.124 1.604 | 3.291 1.388 |
| | 720 | 0.381 0.423 | 0.406 0.441 | 0.462 0.468 | 0.500 0.497 | 0.831 0.657 | 0.463 0.474 | 3.647 1.625 | 3.188 1.540 | 3.257 1.357 |
| | Avg | **0.331 0.380** | 0.354 0.394 | 0.414 0.427 | 0.439 0.452 | 0.559 0.515 | 0.4370.449 | 4.431 1.729 | 2.686 1.494 | 3.095 1.352 |
| Electricity | 96 | 0.132 0.223 | 0.139 0.238 | 0.168 0.222 | 0.187 0.304 | 0.197 0.282 | 0.193 0.308 | 0.274 0.368 | 0.258 0.357 | 0.375 0.437 |
| | 192 | 0.158 0.241 | 0.153 0.251 | 0.184 0.239 | 0.199 0.196 | 0.285 0.201 | 0.315 0.296 | 0.386 0.266 | 0.368 0.348 | 0.442 0.473 |
| | 336 | 0.163 0.260 | 0.169 0.266 | 0.198 0.260 | 0.212 0.329 | 0.209 0.301 | 0.214 0.329 | 0.300 0.394 | 0.280 0.380 | 0.439 0.473 |
| | 720 | 0.199 0.291 | 0.206 0.297 | 0.220 0.300 | 0.233 0.345 | 0.245 0.333 | 0.246 0.355 | 0.373 0.439 | 0.283 0.376 | 0.980 0.814 |
| | Avg | **0.162** 0.253 | 0.167 0.263 | 0.192 **0.245** | 0.208 0.323 | 0.212 0.300 | 0.214 0.327 | 0.311 0.397 | 0.313 0.401 | 0.559 0.549 |
| Traffic | 96 | 0.407 0.282 0 | 0.388 0.282 | 0.593 0.321 | 0.607 0.392 | 0.650 0.396 | 0.587 0.366 | 0.719 0.391 | 0.684 0.384 | 0.843 0.453 |
| | 192 | 0.423 0.287 | 0.407 0.290 | 0.617 0.336 | 0.621 0.399 | 0.598 0.370 | 0.604 0.373 | 0.696 0.379 | 0.685 0.390 | 0.847 0.453 |
| | 336 | 0.430 0.296 | 0.412 0.294 | 0.629 0.336 | 0.622 0.396 | 0.605 0.373 | 0.621 0.383 | 0.777 0.420 | 0.734 0.408 | 0.853 0.455 |
| | 720 | 0.463 0.315 | 0.450 0.312 | 0.640 0.350 | 0.632 0.396 | 0.645 0.394 | 0.626 0.382 | 0.864 0.472 | 0.717 0.396 | 1.500 0.805 |
| | Avg | 0.430 0.295 | **0.414 0.294** | 0.620 0.336 | 0.621 0.396 | 0.625 0.383 | 0.610 0.376 | 0.764 0.416 | 0.705 0.395 | 1.011 0.541 |
| Weather | 96 | 0.150 0.202 | 0.162 0.212 | 0.152 0.220 | 0.197 0.281 | 0.196 0.255 | 0.217 0.296 | 0.300 0.384 | 0.458 0.490 | 0.369 0.406 |
| | 192 | 0.198 0.246 | 0.204 0.248 | 0.209 0.261 | 0.237 0.312 | 0.237 0.296 | 0.276 0.336 | 0.598 0.544 | 0.658 0.589 | 0.416 0.435 |
| | 336 | 0.245 0.286 | 0.254 0.286 | 0.280 0.306 | 0.298 0.353 | 0.283 0.335 | 0.339 0.380 | 0.578 0.521 | 0.797 0.652 | 0.455 0.454 |
| | 720 | 0.324 0.342 | 0.326 0.337 | 0.365 0.359 | 0.352 0.288 | 0.345 0.381 | 0.403 0.428 | 1.059 0.741 | 0.869 0.675 | 0.535 0.520 |
| | Avg | 0.229 0.271 | 0.237 **0.270** | **0.236** 0.287 | 0.271 0.334 | 0.265 0.317 | 0.309 0.360 | 0.634 0.548 | 0.696 0.602 | 0.444 0.454 |
| ILI | 24 | 1.974 0.886 | 2.063 0.881 | 2.317 0.934 | 2.527 1.000 | 2.398 1.040 | 3.228 1.260 | 5.764 1.677 | 4.480 1.444 | 5.914 1.734 |
| | 36 | 2.028 0.976 | 1.868 0.892 | 1.972 0.900 | 2.615 1.007 | 2.646 1.088 | 2.679 1.080 | 4.755 1.467 | 4.799 1.467 | 6.631 1.845 |
| | 48 | 2.353 1.115 | 1.790 0.884 | 2.238 0.900 | 2.359 0.972 | 2.614 1.086 | 2.622 1.078 | 4.763 1.469 | 4.800 1.468 | 6.736 1.857 |
| | 60 | 2.425 1.203 | 1.979 0.957 | 2.027 0.928 | 2.487 1.016 | 2.804 1.146 | 2.857 1.15 | 5.264 1.564 | 5.278 1.560 | 6.870 1.879 |
| | Avg | 2.195 1.045 | **1.925** 0.903 | 2.139 **0.901** | 2.497 1.004 | 2.616 1.090 | 2.847 1.144 | 5.137 1.544 | 4.839 1.485 | 6.538 1.829 |
| $1^{st}$ count | | 5 | 5 | 4 | 0 | 0 | 0 | 0 | 0 | 0 |

Table S7: Long-term Forecasting Results (MSE, MAE). TEST uses GPT2-Medium as the backbone. The past sequence length is set as 36 for ILI and 96 for the others. All the results are averaged from 4 different prediction lengths, that is {24, 36, 48, 60} for ILI and {96, 192, 336, 720} for the others.

### 2.2.3 LONG-TERM FORECASTING

We follow the classical experiment settings and the results of SOTA models in Wu et al. (2023) (ICLR 2023). The results are shown in Table S7. Overall, TEST achieves comparable performance to SOTA models TimesNet and Dlinear, and outperforms other baselines.

### 2.2.4 FEW-SHOT FORECASTING

For the few-shot forecasting task, only 10% percentage timesteps of training data are used, and the other two parts remain unchanged. We follow the classical experiment settings and the results of SOTA models in Zhou et al. (2023) (NeurIPS 2023). The results are shown in Table S8. Overall, TEST has comparable performance with the SOTA baselines PatchTST and Dlinear, and SOTA LLM for TS method GPT4TS.

| Methods | | TEST | GPT4TS | DLinear | PatchTST | TimesNet | FEDformer | Autoformer | Stationary | ETSformer | LightTS | Informer | Reformer |
|---|---|---|---|---|---|---|---|---|---|---|---|---|---|
| Weather | 96 | 0.163 0.213 | 0.163 0.215 | 0.171 0.224 | 0.165 0.215 | 0.184 0.230 | 0.188 0.253 | 0.221 0.297 | 0.192 0.234 | 0.199 0.272 | 0.217 0.269 | 0.374 0.401 | 0.335 0.380 |
| | 192 | 0.230 0.263 | 0.210 0.254 | 0.215 0.263 | 0.210 0.257 | 0.245 0.283 | 0.250 0.304 | 0.270 0.322 | 0.269 0.295 | 0.279 0.332 | 0.259 0.304 | 0.552 0.478 | 0.522 0.462 |
| | 336 | 0.278 0.282 | 0.256 0.292 | 0.258 0.299 | 0.259 0.297 | 0.305 0.321 | 0.312 0.346 | 0.320 0.351 | 0.370 0.357 | 0.356 0.386 | 0.303 0.334 | 0.724 0.541 | 0.715 0.535 |
| | 720 | 0.301 0.328 | 0.321 0.339 | 0.320 0.346 | 0.332 0.346 | 0.381 0.371 | 0.387 0.393 | 0.390 0.396 | 0.441 0.405 | 0.437 0.448 | 0.377 0.382 | 0.739 0.558 | 0.611 0.500 |
| | Avg | 0.243 **0.272** | **0.238** 0.275 | 0.241 0.283 | 0.242 0.279 | 0.279 0.301 | 0.284 0.324 | 0.300 0.342 | 0.318 0.323 | 0.318 0.360 | 0.289 0.322 | 0.597 0.495 | 0.546 0.469 |
| ETTh1 | 96 | 0.455 0.457 | 0.458 0.456 | 0.492 0.495 | 0.516 0.485 | 0.861 0.628 | 0.512 0.499 | 0.613 0.552 | 0.918 0.639 | 1.112 0.806 | 1.298 0.838 | 1.179 0.792 | 1.184 0.790 |
| | 192 | 0.572 0.519 | 0.570 0.516 | 0.565 0.538 | 0.598 0.524 | 0.797 0.593 | 0.624 0.555 | 0.722 0.598 | 0.915 0.629 | 1.155 0.823 | 1.322 0.854 | 1.199 0.806 | 1.295 0.850 |
| | 336 | 0.611 0.531 | 0.608 0.535 | 0.721 0.622 | 0.657 0.550 | 0.941 0.648 | 0.691 0.574 | 0.750 0.619 | 0.939 0.644 | 1.179 0.832 | 1.347 0.870 | 1.202 0.811 | 1.294 0.854 |
| | 720 | 0.723 0.594 | 0.725 0.591 | 0.986 0.743 | 0.762 0.610 | 0.877 0.641 | 0.728 0.614 | 0.721 0.616 | 0.887 0.645 | 1.273 0.874 | 1.534 0.947 | 1.217 0.825 | 1.223 0.838 |
| | Avg | **0.479** 0.525 | 0.590 0.525 | 0.691 0.600 | 0.633 0.542 | 0.869 0.628 | 0.639 0.561 | 0.702 0.596 | 0.915 0.639 | 1.180 0.834 | 1.375 0.877 | 1.199 0.809 | 1.249 0.833 |
| ETTh2 | 96 | 0.332 0.374 | 0.331 0.374 | 0.357 0.411 | 0.353 0.389 | 0.378 0.409 | 0.382 0.416 | 0.413 0.451 | 0.389 0.411 | 0.678 0.619 | 2.022 1.006 | 3.837 1.508 | 3.788 1.533 |
| | 192 | 0.401 0.433 | 0.402 0.411 | 0.569 0.519 | 0.403 0.414 | 0.490 0.467 | 0.478 0.474 | 0.474 0.477 | 0.473 0.455 | 0.785 0.666 | 2.329 1.104 | 3.856 1.513 | 3.552 1.483 |
| | 336 | 0.408 0.440 | 0.406 0.433 | 0.671 0.572 | 0.426 0.441 | 0.537 0.494 | 0.504 0.501 | 0.547 0.543 | 0.507 0.480 | 0.839 0.694 | 2.453 1.122 | 3.952 1.526 | 3.395 1.526 |
| | 720 | 0.459 0.480 | 0.449 0.464 | 0.824 0.648 | 0.477 0.480 | 0.510 0.491 | 0.499 0.509 | 0.516 0.523 | 0.477 0.472 | 1.273 0.874 | 3.816 1.407 | 3.842 1.503 | 3.205 1.401 |
| | Avg | 0.401 0.432 | **0.397 0.421** | 0.605 0.538 | 0.415 0.431 | 0.479 0.465 | 0.466 0.475 | 0.488 0.499 | 0.462 0.455 | 0.894 0.713 | 2.655 1.160 | 3.872 1.513 | 3.485 1.486 |
| ETTm1 | 96 | 0.392 0.401 | 0.390 0.404 | 0.352 0.392 | 0.410 0.419 | 0.583 0.501 | 0.578 0.518 | 0.774 0.614 | 0.761 0.568 | 0.911 0.688 | 0.921 0.682 | 1.162 0.785 | 1.442 0.847 |
| | 192 | 0.423 0.426 | 0.429 0.423 | 0.382 0.412 | 0.437 0.434 | 0.630 0.528 | 0.617 0.546 | 0.754 0.592 | 0.781 0.574 | 0.955 0.703 | 0.957 0.701 | 1.172 0.793 | 1.444 0.862 |
| | 336 | 0.471 0.444 | 0.469 0.439 | 0.419 0.434 | 0.476 0.454 | 0.725 0.568 | 0.998 0.775 | 0.869 0.677 | 0.803 0.587 | 0.991 0.719 | 0.998 0.716 | 1.227 0.908 | 1.450 0.866 |
| | 720 | 0.552 0.501 | 0.569 0.498 | 0.490 0.477 | 0.681 0.556 | 0.769 0.549 | 0.693 0.579 | 0.810 0.630 | 0.844 0.581 | 1.062 0.747 | 1.007 0.719 | 1.207 0.797 | 1.366 0.850 |
| | Avg | 0.574 0.443 | 0.464 0.441 | **0.411 0.429** | 0.501 0.466 | 0.677 0.537 | 0.722 0.605 | 0.802 0.628 | 0.797 0.578 | 0.980 0.714 | 0.971 0.705 | 1.192 0.821 | 1.426 0.856 |
| ETTm2 | 96 | 0.233 0.262 | 0.188 0.269 | 0.213 0.303 | 0.191 0.274 | 0.212 0.285 | 0.291 0.399 | 0.352 0.454 | 0.229 0.308 | 0.331 0.430 | 0.813 0.688 | 3.203 1.407 | 4.195 1.628 |
| | 192 | 0.303 0.302 | 0.251 0.309 | 0.278 0.345 | 0.252 0.317 | 0.270 0.323 | 0.307 0.379 | 0.694 0.691 | 0.291 0.343 | 0.400 0.464 | 1.008 0.768 | 3.112 1.387 | 4.042 1.601 |
| | 336 | 0.359 0.341 | 0.307 0.346 | 0.338 0.385 | 0.306 0.353 | 0.323 0.353 | 0.543 0.559 | 2.408 1.407 | 0.348 0.376 | 0.469 0.498 | 1.031 0.775 | 3.255 1.421 | 3.963 1.585 |
| | 720 | 0.452 0.419 | 0.426 0.417 | 0.436 0.440 | 0.433 0.427 | 0.474 0.449 | 0.712 0.614 | 1.913 1.166 | 0.461 0.438 | 0.589 0.557 | 1.096 0.791 | 3.909 1.543 | 3.711 1.532 |
| | Avg | 0.317 **0.309** | **0.293** 0.335 | 0.316 0.368 | 0.296 0.343 | 0.320 0.353 | 0.463 0.488 | 1.342 0.930 | 0.332 0.366 | 0.447 0.487 | 0.987 0.756 | 3.370 1.440 | 3.978 1.587 |
| Electricity | 96 | 0.143 0.235 | 0.139 0.237 | 0.150 0.253 | 0.140 0.238 | 0.299 0.373 | 0.231 0.323 | 0.261 0.348 | 0.420 0.466 | 0.599 0.587 | 0.350 0.425 | 1.259 0.919 | 0.993 0.784 |
| | 192 | 0.158 0.255 | 0.156 0.252 | 0.164 0.264 | 0.160 0.255 | 0.305 0.379 | 0.261 0.356 | 0.338 0.406 | 0.411 0.459 | 0.620 0.598 | 0.376 0.448 | 1.160 0.873 | 0.938 0.753 |
| | 336 | 0.176 0.275 | 0.175 0.270 | 0.181 0.282 | 0.180 0.276 | 0.319 0.391 | 0.360 0.445 | 0.410 0.474 | 0.434 0.473 | 0.662 0.619 | 0.428 0.485 | 1.157 0.872 | 0.925 0.745 |
| | 720 | 0.230 0.311 | 0.233 0.317 | 0.223 0.321 | 0.241 0.323 | 0.369 0.426 | 0.530 0.585 | 0.715 0.685 | 0.510 0.521 | 0.757 0.664 | 0.611 0.597 | 1.203 0.898 | 1.004 0.790 |
| | Avg | 0.176 0.269 | 0.176 0.269 | 0.180 0.280 | 0.180 0.273 | 0.323 0.392 | 0.346 0.427 | 0.431 0.478 | 0.444 0.480 | 0.660 0.617 | 0.441 0.489 | 1.195 0.891 | 0.965 0.768 |
| Traffic | 96 | 0.415 0.317 | 0.414 0.297 | 0.419 0.298 | 0.403 0.289 | 0.719 0.416 | 0.639 0.400 | 0.672 0.405 | 1.412 0.802 | 1.643 0.855 | 1.157 0.636 | 1.557 0.821 | 1.527 0.815 |
| | 192 | 0.425 0.300 | 0.426 0.301 | 0.434 0.305 | 0.415 0.296 | 0.748 0.428 | 0.637 0.416 | 0.727 0.424 | 1.419 0.806 | 1.641 0.854 | 1.207 0.661 | 1.454 0.765 | 1.538 0.817 |
| | 336 | 0.436 0.310 | 0.434 0.303 | 0.449 0.313 | 0.426 0.304 | 0.853 0.471 | 0.655 0.427 | 0.749 0.454 | 1.443 0.815 | 1.711 0.878 | 1.334 0.713 | 1.521 0.812 | 1.550 0.819 |
| | 720 | 0.489 0.338 | 0.487 0.337 | 0.484 0.336 | 0.474 0.331 | 1.485 0.825 | 0.722 0.456 | 0.847 0.499 | 1.539 0.837 | 2.660 1.157 | 1.292 0.726 | 1.605 0.846 | 1.588 0.833 |
| | Avg | 0.441 0.316 | 0.440 0.310 | 0.447 0.313 | **0.430 0.305** | 0.951 0.535 | 0.663 0.425 | 0.749 0.446 | 1.453 0.815 | 1.914 0.936 | 1.248 0.684 | 1.534 0.811 | 1.551 0.821 |
| $1^{st}$ count | | 5 | 5 | 4 | 0 | 0 | 0 | 0 | 0 | 0 | 0 | 0 | 0 |

Table S8: Few-shot Forecasting Results (MSE, MAE). TEST uses GPT2-Medium as the backbone. All the results are averaged from 4 different prediction lengths, that is {96, 192, 336, 720}.

### 2.2.5 ZERO-SHOT FORECASTING

Zero-shot Forecasting task can evaluate the cross datasets adaption ability. Which means that the method is evaluated to perform on a dataset (without any training data from this dataset) when it is trained from another dataset. The results are summarized in Table S9. TEST outperforms all recent SOTA methods. TEST is comparable to N-BEATS without any meta-learning design and GPT4TS.

| Methods Metric | M4 sMAPE | M3 sMAPE | TOURISM MAPE | ELECTR ND×100 | Average | $1^{st}$ count |
|---|---|---|---|---|---|---|
| N-BEATS | **11.70** | **12.44** | 18.82 | 17.8 | 15.19 | 2 |
| DLinear | 15.33 | 14.03 | 28.51 | 17.6 | 18.86 | 0 |
| TimesNet | 13.55 | 14.17 | 28.84 | 19.3 | 18.96 | 0 |
| PatchTST | 13.22 | 13.06 | 27.10 | 17.3 | 17.67 | 0 |
| ETSformer | 27.74 | 16.03 | 180.40 | 44.2 | 67.09 | 0 |
| LightTS | 13.62 | 17.90 | 66.99 | 19.6 | 29.52 | 0 |
| Stationary | 13.32 | 15.29 | 43.75 | 22.0 | 23.59 | 0 |
| FEDformer | 15.04 | 13.53 | 31.55 | 18.4 | 19.63 | 0 |
| Autoformer | 20.02 | 15.87 | 40.39 | 33.9 | 27.54 | 0 |
| Informer | 19.04 | 15.82 | 35.82 | 21.2 | 22.97 | 0 |
| Reformer | 14.09 | 13.37 | 25.48 | 21.6 | 18.63 | 0 |
| GPT2(6) | 13.12 | 13.06 | 22.14 | **17.2** | 16.38 | 1 |
| TEST | 13.10 | 12.56 | **18.17** | 17.9 | 15.93 | 1 |

Table S9: Zero-shot learning results. Dataset-specific metrics aggregated over each dataset. A lower value indicates better performance. The source dataset of M3, Tourism, Electricity are M4. For M4, the source data for N-BEATS is FRED, and M3 for other models.

## 2.3 CLASSIFICATION TASKS

All the deep learning networks are implemented in PyTorch and trained on NVIDIA V100 32GB GPUs. We use Area Under Curve of Receiver Operating Characteristic (AUC-ROC) as metrics. Meanwhile, we compute the average rank, the number of Top-1, Top-3, and Top-5 accuracy to show the robustness of different methods. All experiments are repeated 3 times and the mean of the metrics is used in the final results.

### 2.3.1 DATASET DETAILS

We present accuracy scores for all 30 kinds of multivariate TS datasets in UEA archive Bagnall et al. (2018). UEA consists of 30 different datasets. Details of these datasets are shown in Table S10

| Dataset | Train Cases | Test Cases | Dimensions | Length | Classes |
|---|---|---|---|---|---|
| ArticularyWordRecognition | 275 | 30 | 9 | 144 | 25 |
| AtrialFibrillation | 15 | 15 | 2 | 640 | 3 |
| BasicMotions | 40 | 40 | 4 | 100 | 4 |
| CharacterTrajectories | 1422 | 1436 | 3 | 182 | 20 |
| Cricket | 108 | 72 | 6 | 17984 | 5 |
| DuckDuckGeese | 60 | 40 | 1345 | 270 | 5 |
| EigenWorms | 128 | 131 | 6 | 17984 | 5 |
| Epilepsy | 137 | 138 | 3 | 206 | 4 |
| EthanolConcentration | 261 | 263 | 3 | 1751 | 4 |
| ERing | 30 | 20 | 4 | 65 | 6 |
| FaceDetection | 5890 | 3524 | 144 | 62 | 2 |
| FingerMovements | 316 | 100 | 28 | 50 | 2 |
| HandMovementDirection | 320 | 147 | 10 | 400 | 4 |
| Handwriting | 150 | 850 | 3 | 152 | 26 |
| Heartbeat | 204 | 105 | 61 | 495 | 2 |
| JapaneseVowels | 270 | 370 | 12 | 29 | 9 |
| Libras | 180 | 280 | 2 | 45 | 15 |
| LSST | 2459 | 2466 | 6 | 36 | 14 |
| InsectWingbeat | 30000 | 20000 | 200 | 78 | 10 |
| MotorImagery | 278 | 100 | 64 | 3000 | 2 |
| NATOPS | 180 | 180 | 24 | 51 | 6 |
| PenDigits | 7494 | 3498 | 2 | 8 | 10 |
| PEMS-SF | 267 | 173 | 963 | 144 | 7 |
| Phoneme | 3315 | 3353 | 11 | 217 | 39 |
| RacketSports | 151 | 152 | 6 | 30 | 4 |
| SelfRegulationSCP1 | 268 | 293 | 6 | 896 | 2 |
| SelfRegulationSCP2 | 200 | 180 | 7 | 1152 | 2 |
| SpokenArabicDigits | 6599 | 2199 | 13 | 93 | 10 |
| StandWalkJump | 12 | 15 | 4 | 2500 | 3 |
| UWaveGestureLibrary | 120 | 320 | 3 | 315 | 8 |

Table S10: UEA Classification Dataset Details

### 2.3.2 BASELINE DETAILS

For classification, we refer to the SOTA methods: Three benchmarks Bostrom et al. (2018) (EDI, DTWI, and DTWD) are based on Euclidean Distance, dimension-independent dynamic time warping, and dimension-dependent dynamic time warping; MLSTM-FCNs Karim et al. (2019) applies an LSTM layer and stacked CNN layers to generate features; WEASEL-MUSE Schäfer & Leser (2017) is a bag-of-pattern based approach which extracts and represents features to words. Scalable Representation Learning (SRL) Franceschi et al. (2019a) employs negative sampling techniques with an encoder-based architecture to learn the representation; TapNet Zhang et al. (2020) is a recent model with an attentional prototype learning in its deep learning-based network; ShapeNet Li et al. (2021a) projects the subsequences into a unified space and applies clustering to find the shapelets; Rocket and MiniRocket Dempster et al. (2021) use random convolutional kernels to extract features from univariate time series; RL-PAM Gao et al. (2022) introduces reinforcement learning to the pattern mining; TStamp Transformer Zerveas et al. (2021) takes the values at each timestamp as the input for a transformer encoder; SVP-T Zuo et al. (2023) uses differnt variables and positions (time interval) as the inputs (shape-level).

### 2.3.3 MULTIVARIATE TIME SERIES CLASSIFICATION

We follow the classical experiment settings in multivariate time series classification tasks Bostrom et al. (2018). The results are shown in Table S11. Overall, TEST achieves comparable performance to SOTA models and outperforms most baselines.

| | EDI | DTWI | DTWD | MLSTM-FCNs | WEASEL+MUSE | SRL | TapNet | ShapeNet | Rocket | MiniRocket | RLPAM | TStamp | SVP-T | TEST |
|---|---|---|---|---|---|---|---|---|---|---|---|---|---|---|
| AWR | 0.970 | 0.980 | 0.987 | 0.973 | 0.990 | 0.987 | 0.987 | 0.987 | 0.996 | 0.992 | 0.923 | 0.983 | 0.993 | 0.994 |
| AF | 0.267 | 0.267 | 0.220 | 0.267 | 0.333 | 0.133 | 0.333 | 0.400 | 0.249 | 0.133 | 0.733 | 0.200 | 0.400 | 0.420 |
| BM | 0.676 | 1.000 | 0.975 | 0.950 | 1.000 | 1.000 | 1.000 | 1.000 | 0.990 | 1.000 | 1.000 | 0.975 | 1.000 | 1.000 |
| CT | 0.964 | 0.969 | 0.989 | 0.985 | 0.990 | 0.994 | 0.997 | 0.980 | N/A | 0.993 | 0.978 | N/A | 0.990 | 0.989 |
| CK | 0.944 | 0.986 | 1.000 | 0.917 | 1.000 | 0.986 | 0.958 | 0.986 | 1.000 | 0.986 | 1.000 | 0.958 | 1.000 | 1.000 |
| DDG | 0.275 | 0.550 | 0.600 | 0.675 | 0.575 | 0.675 | 0.575 | 0.725 | 0.461 | 0.650 | 0.700 | 0.480 | 0.700 | 0.675 |
| EW | 0.549 | N/A | 0.618 | 0.504 | 0.890 | 0.878 | 0.489 | 0.878 | 0.863 | 0.962 | 0.908 | N/A | 0.923 | 0.878 |
| EP | 0.666 | 0.978 | 0.964 | 0.761 | 1.000 | 0.957 | 0.971 | 0.987 | 0.991 | 1.000 | 0.978 | 0.920 | 0.986 | 0.985 |
| ER | 0.133 | 0.914 | 0.929 | 0.133 | 0.133 | 0.133 | 0.133 | 0.133 | 0.981 | 0.981 | 0.819 | 0.933 | 0.937 | 0.937 |
| EC | 0.293 | 0.304 | 0.323 | 0.373 | 0.430 | 0.236 | 0.323 | 0.312 | 0.447 | 0.468 | 0.369 | 0.337 | 0.331 | 0.373 |
| FD | 0.519 | 0.000 | 0.529 | 0.545 | 0.545 | 0.528 | 0.556 | 0.602 | 0.694 | 0.620 | 0.621 | 0.681 | 0.512 | 0.512 |
| FM | 0.550 | 0.520 | 0.530 | 0.580 | 0.490 | 0.540 | 0.530 | 0.580 | 0.553 | 0.550 | 0.640 | 0.776 | 0.600 | 0.770 |
| HMD | 0.278 | 0.306 | 0.231 | 0.365 | 0.365 | 0.270 | 0.378 | 0.338 | 0.446 | 0.392 | 0.635 | 0.608 | 0.392 | 0.444 |
| HW | 0.200 | 0.316 | 0.286 | 0.286 | 0.605 | 0.533 | 0.357 | 0.452 | 0.567 | 0.507 | 0.522 | 0.305 | 0.433 | 0.431 |
| HB | 0.619 | 0.658 | 0.717 | 0.663 | 0.727 | 0.737 | 0.751 | 0.756 | 0.718 | 0.771 | 0.779 | 0.712 | 0.790 | 0.791 |
| IW | 0.128 | N/A | N/A | 0.167 | N/A | 0.160 | 0.208 | 0.250 | N/A | 0.595 | 0.352 | 0.684 | 0.184 | 0.572 |
| JV | 0.924 | 0.959 | 0.949 | 0.976 | 0.973 | 0.989 | 0.965 | 0.984 | 0.965 | 0.989 | 0.935 | 0.994 | 0.978 | 0.991 |
| LB | 0.833 | 0.894 | 0.870 | 0.856 | 0.878 | 0.867 | 0.850 | 0.856 | 0.906 | 0.922 | 0.794 | 0.844 | 0.883 | 0.884 |
| LSST | 0.456 | 0.575 | 0.551 | 0.373 | 0.590 | 0.558 | 0.568 | 0.590 | 0.632 | 0.643 | 0.643 | 0.381 | 0.666 | 0.595 |
| MI | 0.510 | N/A | 0.500 | 0.510 | 0.500 | 0.540 | 0.590 | 0.610 | 0.531 | 0.550 | 0.610 | N/A | 0.650 | 0.650 |
| NT | 0.850 | 0.850 | 0.883 | 0.889 | 0.870 | 0.944 | 0.939 | 0.883 | 0.885 | 0.928 | 0.950 | 0.900 | 0.906 | 0.902 |
| PD | 0.705 | 0.939 | 0.977 | 0.978 | 0.948 | 0.983 | 0.980 | 0.977 | 0.996 | N/A | 0.982 | 0.974 | 0.983 | 0.979 |
| PM | 0.973 | 0.734 | 0.711 | 0.699 | 0.000 | 0.688 | 0.751 | 0.751 | 0.856 | 0.522 | 0.632 | 0.919 | 0.867 | 0.860 |
| PH | 0.104 | 0.151 | 0.151 | 0.110 | 0.190 | 0.246 | 0.175 | 0.298 | 0.284 | 0.292 | 0.175 | 0.088 | 0.176 | 0.196 |
| RS | 0.868 | 0.842 | 0.803 | 0.803 | 0.934 | 0.862 | 0.868 | 0.882 | 0.928 | 0.868 | 0.868 | 0.829 | 0.842 | 0.851 |
| SCP1 | 0.771 | 0.765 | 0.775 | 0.874 | 0.710 | 0.846 | 0.652 | 0.782 | 0.866 | 0.925 | 0.802 | 0.925 | 0.884 | 0.870 |
| SCP2 | 0.483 | 0.533 | 0.539 | 0.472 | 0.460 | 0.556 | 0.550 | 0.578 | 0.514 | 0.522 | 0.632 | 0.589 | 0.600 | 0.579 |
| SAD | 0.967 | 0.959 | 0.963 | 0.990 | 0.982 | 0.956 | 0.983 | 0.975 | 0.630 | 0.620 | 0.621 | 0.993 | 0.986 | 0.982 |
| SWJ | 0.200 | 0.333 | 0.200 | 0.067 | 0.333 | 0.400 | 0.400 | 0.533 | 0.456 | 0.333 | 0.667 | 0.267 | 0.467 | 0.468 |
| UGL | 0.881 | 0.868 | 0.903 | 0.891 | 0.916 | 0.884 | 0.894 | 0.906 | 0.944 | 0.938 | 0.944 | 0.903 | 0.941 | 0.933 |
| Avg.Rank | 10.933 | 9.480 | 8.821 | 8.756 | 6.890 | 7.120 | 6.956 | 5.523 | 5.423 | 5.013 | 5.059 | 7.484 | 4.032 | 4.012 |
| Num.Top-1 | 1 | 1 | 1 | 0 | 5 | 1 | 2 | 3 | 5 | 5 | 6 | 4 | 4 | 6 |
| Num.Top-3 | 1 | 2 | 1 | 1 | 6 | 6 | 3 | 7 | 12 | 14 | 16 | 9 | 17 | 18 |
| Num.Top-5 | 2 | 2 | 3 | 5 | 15 | 12 | 13 | 17 | 16 | 20 | 19 | 10 | 23 | 24 |
| P-value | 0.000 | 0.000 | 0.000 | 0.000 | 0.006 | 0.003 | 0.000 | 0.118 | 0.217 | 0.765 | 0.967 | 0.047 | 0.044 | 0.040 |

Table S11: Accuracies on All Datasets of the UEA Archive

### 2.4 REPRESENTATION TASKS

We assess the quality of our learned representations on supervised tasks in a standard manner by using them for time series classification Franceschi et al. (2019b). All the deep learning networks

are implemented in PyTorch and trained on NVIDIA V100 32GB GPUs. We use Area Under Curve of Receiver Operating Characteristic (AUC-ROC) as metrics.

### 2.4.1 DATASET DETAILS

We represent the results for all 128 kinds of univariate TS datasets in UCR archive Dau et al. (2019), which is a standard set of varied univariate datasets.

### 2.4.2 BASELINE DETAILS

The compared method includes SOTAs of unsupervised time series representation: T-Loss Franceschi et al. (2019b), TS-TCC Eldele et al. (2021b), TST Zerveas et al. (2021) and TNC Tonekaboni et al. (2021), TS2Vec Yue et al. (2022).

### 2.4.3 CLASSIFICATION BASED ON REPRESENTATION

We assess the quality of our learned representations on supervised tasks in a standard manner by using them for time series classification Franceschi et al. (2019b). In this setting, we show that our method outperforms SOTA unsupervised methods, and notably achieves performance close to the supervised SOTA method as shown in Table S12.

For each considered dataset with a train / test split, we unsupervisedly train an encoder using its train set. We then train an SVM with radial basis function kernel on top of the learned features using the train labels of the dataset, and output the corresponding classification score on the test set.

|  | TEST | TCN | TS2Vec | T-Loss | TNC |
|---|---|---|---|---|---|
| Adiac | 0.776 | 0.768 | 0.765 | 0.675 | 0.726 |
| ArrowHead | 0.825 | 0.857 | 0.817 | 0.766 | 0.703 |
| Beef | 0.766 | 0.768 | 0.633 | 0.667 | 0.733 |
| BeetleFly | 0.853 | 0.900 | 0.900 | 0.800 | 0.850 |
| BirdChicken | 0.808 | 0.803 | 0.800 | 0.850 | 0.750 |
| Car | 0.883 | 0.834 | 0.700 | 0.833 | 0.683 |
| CBF | 1.000 | 1.000 | 1.000 | 0.983 | 0.983 |
| ChlorineConcentration | 0.810 | 0.832 | 0.812 | 0.749 | 0.760 |
| CinCECGTorso | 0.815 | 0.829 | 0.825 | 0.713 | 0.669 |
| Coffee | 1.000 | 1.000 | 1.000 | 1.000 | 1.000 |
| Computers | 0.632 | 0.660 | 0.660 | 0.664 | 0.684 |
| CricketX | 0.802 | 0.787 | 0.805 | 0.713 | 0.623 |
| CricketY | 0.754 | 0.749 | 0.769 | 0.728 | 0.597 |
| CricketZ | 0.787 | 0.794 | 0.790 | 0.708 | 0.682 |
| DiatomSizeReduction | 0.980 | 0.985 | 0.987 | 0.984 | 0.993 |
| DistalPhalanxOutlineCorrect | 0.776 | 0.761 | 0.757 | 0.775 | 0.754 |
| DistalPhalanxOutlineAgeGroup | 0.714 | 0.727 | 0.719 | 0.727 | 0.741 |
| DistalPhalanxTW | 0.662 | 0.698 | 0.683 | 0.676 | 0.669 |
| Earthquakes | 0.746 | 0.748 | 0.748 | 0.748 | 0.748 |
| ECG200 | 0.893 | 0.920 | 0.880 | 0.940 | 0.830 |
| ECG5000 | 0.935 | 0.935 | 0.934 | 0.933 | 0.937 |
| ECGFiveDays | 1.000 | 1.000 | 1.000 | 1.000 | 0.999 |
| ElectricDevices | 0.714 | 0.721 | 0.719 | 0.707 | 0.700 |
| FaceAll | 0.789 | 0.771 | 0.805 | 0.786 | 0.766 |
| FaceFour | 0.834 | 0.932 | 0.932 | 0.920 | 0.659 |
| FacesUCR | 0.939 | 0.924 | 0.926 | 0.884 | 0.789 |
| FiftyWords | 0.781 | 0.771 | 0.774 | 0.732 | 0.653 |
| Fish | 0.937 | 0.926 | 0.937 | 0.891 | 0.817 |
| FordA | 0.940 | 0.936 | 0.948 | 0.928 | 0.902 |
| FordB | 0.789 | 0.794 | 0.807 | 0.793 | 0.733 |
| GunPoint | 0.983 | 0.980 | 0.987 | 0.980 | 0.967 |
| Ham | 0.714 | 0.714 | 0.724 | 0.724 | 0.752 |
| HandOutlines | 0.918 | 0.925 | 0.930 | 0.922 | 0.930 |
| Haptics | 0.510 | 0.526 | 0.536 | 0.490 | 0.474 |
| Herring | 0.625 | 0.644 | 0.609 | 0.594 | 0.594 |
| InlineSkate | 0.389 | 0.418 | 0.407 | 0.371 | 0.378 |
| InsectWingbeatSound | 0.620 | 0.630 | 0.624 | 0.597 | 0.549 |
| ItalyPowerDemand | 0.969 | 0.925 | 0.960 | 0.954 | 0.928 |
| LargeKitchenAppliances0 | 0.855 | 0.845 | 0.875 | 0.789 | 0.776 |
| Lightning2 | 0.846 | 0.869 | 0.820 | 0.869 | 0.869 |
| Lightning7 | 0.866 | 0.863 | 0.822 | 0.795 | 0.767 |
| Mallat | 0.915 | 0.944 | 0.873 | 0.951 | 0.871 |
| Meat | 0.950 | 0.952 | 0.967 | 0.950 | 0.917 |
| MedicalImages | 0.792 | 0.789 | 0.793 | 0.750 | 0.754 |
| MiddlePhalanxOutlineCorrect | 0.811 | 0.838 | 0.825 | 0.825 | 0.818 |

| | | | | | |
|---|---|---|---|---|---|
| MiddlePhalanxOutlineAgeGroup | 0.636 | 0.636 | 0.630 | 0.656 | 0.643 |
| MiddlePhalanxTW | 0.591 | 0.584 | 0.578 | 0.591 | 0.571 |
| MoteStrain | 0.857 | 0.861 | 0.863 | 0.851 | 0.825 |
| NonInvasiveFetalECGThorax1 | 0.923 | 0.930 | 0.919 | 0.878 | 0.898 |
| NonInvasiveFetalECGThorax2 | 0.940 | 0.938 | 0.935 | 0.919 | 0.912 |
| OliveOil | 0.903 | 0.901 | 0.940 | 0.867 | 0.833 |
| OSULeaf | 0.872 | 0.851 | 0.843 | 0.760 | 0.723 |
| PhalangesOutlinesCorrect | 0.794 | 0.809 | 0.823 | 0.784 | 0.787 |
| Phoneme | 0.296 | 0.312 | 0.309 | 0.276 | 0.180 |
| Plane | 1.000 | 1.000 | 0.990 | 0.990 | 1.000 |
| ProximalPhalanxOutlineCorrect | 0.876 | 0.887 | 0.900 | 0.859 | 0.866 |
| ProximalPhalanxOutlineAgeGroup | 0.844 | 0.837 | 0.829 | 0.844 | 0.854 |
| ProximalPhalanxTW | 0.785 | 0.824 | 0.805 | 0.771 | 0.810 |
| RefrigerationDevices | 0.587 | 0.586 | 0.589 | 0.515 | 0.565 |
| ScreenType | 0.405 | 0.414 | 0.397 | 0.416 | 0.509 |
| ShapeletSim | 0.989 | 1.000 | 0.994 | 0.672 | 0.589 |
| ShapesAll | 0.897 | 0.902 | 0.905 | 0.848 | 0.788 |
| SmallKitchenAppliances | 0.723 | 0.731 | 0.733 | 0.677 | 0.725 |
| SonyAIBORobotSurface1 | 0.874 | 0.903 | 0.900 | 0.902 | 0.804 |
| SonyAIBORobotSurface2 | 0.893 | 0.871 | 0.889 | 0.889 | 0.834 |
| StarLightCurves | 0.970 | 0.968 | 0.971 | 0.964 | 0.968 |
| Strawberry | 0.962 | 0.966 | 0.965 | 0.954 | 0.951 |
| SwedishLeaf | 0.939 | 0.945 | 0.942 | 0.914 | 0.880 |
| Symbols | 0.973 | 0.977 | 0.972 | 0.963 | 0.885 |
| SyntheticControl | 0.997 | 0.997 | 0.993 | 0.987 | 1.000 |
| ToeSegmentation1 | 0.933 | 0.917 | 0.947 | 0.939 | 0.864 |
| ToeSegmentation2 | 0.915 | 0.899 | 0.900 | 0.900 | 0.831 |
| Trace | 1.000 | 1.000 | 1.000 | 0.990 | 1.000 |
| TwoLeadECG | 0.982 | 0.986 | 0.987 | 0.999 | 0.993 |
| TwoPatterns | 1.000 | 1.000 | 1.000 | 0.999 | 1.000 |
| UWaveGestureLibraryX | 0.810 | 0.795 | 0.801 | 0.785 | 0.781 |
| UWaveGestureLibraryY | 0.729 | 0.719 | 0.720 | 0.710 | 0.697 |
| UWaveGestureLibraryZ | 0.761 | 0.774 | 0.768 | 0.757 | 0.721 |
| UWaveGestureLibraryAll | 0.935 | 0.930 | 0.934 | 0.896 | 0.903 |
| Wafer | 0.995 | 0.998 | 0.998 | 0.992 | 0.994 |
| Wine | 0.788 | 0.880 | 0.889 | 0.815 | 0.759 |
| WordSynonyms | 0.699 | 0.679 | 0.704 | 0.691 | 0.630 |
| Worms | 0.704 | 0.701 | 0.701 | 0.727 | 0.623 |
| WormsTwoClass | 0.805 | 0.806 | 0.753 | 0.792 | 0.727 |
| Yoga | 0.883 | 0.883 | 0.877 | 0.837 | 0.812 |
| ACSF1 | 0.849 | 0.910 | 0.910 | 0.900 | 0.730 |
| AllGestureWiimoteX | 0.744 | 0.777 | 0.751 | 0.763 | 0.703 |
| AllGestureWiimoteY | 0.754 | 0.796 | 0.774 | 0.726 | 0.699 |
| AllGestureWiimoteZ | 0.744 | 0.749 | 0.770 | 0.723 | 0.646 |
| BME | 0.979 | 0.992 | 0.980 | 0.993 | 0.973 |
| Chinatown | 0.969 | 0.964 | 0.959 | 0.951 | 0.977 |
| Crop | 0.753 | 0.754 | 0.758 | 0.722 | 0.738 |
| EOGHorizontalSignal | 0.544 | 0.569 | 0.522 | 0.605 | 0.442 |
| EOGVerticalSignal | 0.467 | 0.503 | 0.472 | 0.434 | 0.392 |
| EthanolLevel | 0.480 | 0.468 | 0.484 | 0.382 | 0.424 |
| FreezerRegularTrain | 0.983 | 0.996 | 0.983 | 0.956 | 0.991 |
| FreezerSmallTrain | 0.893 | 0.875 | 0.872 | 0.933 | 0.982 |
| Fungi | 0.967 | 0.958 | 0.946 | 1.000 | 0.527 |
| GestureMidAirD1 | 0.637 | 0.608 | 0.615 | 0.608 | 0.431 |
| GestureMidAirD2 | 0.508 | 0.479 | 0.515 | 0.546 | 0.362 |
| GestureMidAirD3 | 0.346 | 0.492 | 0.300 | 0.285 | 0.292 |
| GesturePebbleZ1 | 0.878 | 0.930 | 0.884 | 0.919 | 0.378 |
| GesturePebbleZ2 | 0.842 | 0.873 | 0.848 | 0.899 | 0.316 |
| GunPointAgeSpan | 0.994 | 0.987 | 0.968 | 0.994 | 0.984 |
| GunPointMaleVersusFemale | 1.000 | 1.000 | 1.000 | 0.997 | 0.994 |
| GunPointOldVersusYoung | 1.000 | 1.000 | 1.000 | 1.000 | 1.000 |
| HouseTwenty | 0.944 | 0.917 | 0.941 | 0.933 | 0.782 |
| InsectEPGRegularTrain | 1.000 | 1.000 | 1.000 | 1.000 | 1.000 |
| InsectEPGSmallTrain | 1.000 | 1.000 | 1.000 | 1.000 | 1.000 |
| MelbournePedestrian | 0.954 | 0.959 | 0.956 | 0.944 | 0.942 |
| MixedShapesRegularTrain | 0.915 | 0.917 | 0.922 | 0.905 | 0.911 |
| MixedShapesSmallTrain | 0.884 | 0.861 | 0.856 | 0.860 | 0.813 |
| PickupGestureWiimoteZ | 0.800 | 0.823 | 0.760 | 0.740 | 0.620 |
| PigAirwayPressure | 0.524 | 0.630 | 0.683 | 0.510 | 0.413 |
| PigArtPressure | 0.962 | 0.966 | 0.966 | 0.928 | 0.808 |
| PigCVP | 0.803 | 0.815 | 0.870 | 0.788 | 0.649 |
| PLAID | 0.551 | 0.561 | 0.549 | 0.555 | 0.495 |
| PowerCons | 0.967 | 0.961 | 0.972 | 0.900 | 0.933 |
| Rock | 0.660 | 0.700 | 0.700 | 0.580 | 0.580 |
| SemgHandGenderCh2 | 0.952 | 0.963 | 0.962 | 0.890 | 0.882 |
| SemgHandSubjectCh2 | 0.897 | 0.860 | 0.891 | 0.789 | 0.593 |
| SemgHandMovementCh2 | 0.944 | 0.952 | 0.942 | 0.920 | 0.820 |
| SmoothSubspace | 0.967 | 0.980 | 0.993 | 0.960 | 0.913 |
| UMD | 1.000 | 1.000 | 0.993 | 0.993 | 0.993 |

| | | | | | 0.826 | 0.832 | 0.827 | 0.806 | 0.761 |

Table S12: Accuracies on All Datasets of the UCR Archive

## 2.5 ABLATION

`TEST` contains two contrastive learning strategies: instance-wise contrast and feature-wise contrast, and can use different text embedding vectors as prototypes, we show the impact of these strategies.

### 2.5.1 CONTRASTIVE LEARNING STRATEGIES

As shown in Table S13 and S14, both two contrastive learning strategies can increase the accuracy.

| | ETTm1 | ETTm2 | ETTh1 | ETTh2 | Electricity | Traffic | Weather | ILI |
|---|---|---|---|---|---|---|---|---|
| Instance-wise | 0.621 0.550 | 0.755 0.630 | 0.493 0.453 | 0.580 0.612 | 0.293 0.396 | 0.788 0.620 | 0.463 0.349 | 3.301 4.535 |
| Feature-wise | 0.741 0.559 | 0.793 0.634 | 0.699 0.493 | 0.585 0.628 | 0.286 0.390 | 0.821 0.629 | 0.453 0.388 | 3.139 5.931 |
| TEST | 0.353 0.382 | 0.293 0.334 | 0.414 0.431 | 0.331 0.380 | 0.162 0.253 | 0.430 0.295 | 0.229 0.271 | 2.195 1.045 |

Table S13: Long-term Forecasting Results (MSE, MAE). `TEST` uses different contrastive learning stragegy. All the results are averaged from 4 different prediction lengths, that is {24, 36, 48, 60} for ILI and {96, 192, 336, 720} for the others. The results are average.

| | TEST | Instance-wise | Feature-wise | TimesNet | N-BEATS | ETSformer | DLinear | FEDformer | Stationary | Autoformer | Informer | Reformer |
|---|---|---|---|---|---|---|---|---|---|---|---|---|
| SMAPE | 11.927 | 13.525 | 16.987 | 11.829 | 11.851 | 14.718 | 13.639 | 12.840 | 12.780 | 12.909 | 14.086 | 18.200 |
| MASE | 1.613 | 2.111 | 3.265 | 1.585 | 1.599 | 2.408 | 2.095 | 1.701 | 1.756 | 1.771 | 3.010 | 4.223 |
| OWA | 0.861 | 1.051 | 1.480 | 0.851 | 0.855 | 1.172 | 1.051 | 0.918 | 0.930 | 0.939 | 1.230 | 1.775 |

Table S14: Short-term Forecasting Task on M4. The prediction lengths are in [6, 48] and results are averaged from several datasets.

### 2.5.2 TEXT PROTOTYPES

The number and the type of text prototypes will lead to different results.

As shown in Table S15. We randomly select 1, 2, 4, 6, 8, 10, 12, 14, 16, 18, 20, 22 prototypes. The accuracy and number are basically positively correlated. The results of 10 prototypes are almost optimal.

As shown in Table S16. We randomly select 10 prototypes 10 times. The accuracy is basically consistent. Therefore, the type of prototypes has almost no impact on the results.

| | 1 | 2 | 4 | 6 | 8 | 10 | 12 | 14 | 16 | 18 | 20 | 22 |
|---|---|---|---|---|---|---|---|---|---|---|---|---|
| SMAPE | 30.901 | 20.201 | 17.415 | 16.997 | 13.820 | 11.927 | 11.710 | 11.638 | 11.094 | 11.098 | 10.953 | 10.885 |
| MASE | 6.590 | 4.515 | 3.910 | 3.595 | 2.580 | 1.613 | 1.408 | 1.195 | 1.301 | 1.306 | 1.471 | 1.310 |
| OWA | 3.779 | 2.050 | 1.451 | 1.484 | 0.990 | 0.861 | 0.872 | 0.801 | 0.910 | 0.902 | 0.838 | 0.830 |

Table S15: Short-term Forecasting Task on M4. The results are reported with different number of text prototypes.

| | 1 | 2 | 3 | 4 | 5 | 6 | 7 | 8 | 9 | 10 | Avg. | Std. |
|---|---|---|---|---|---|---|---|---|---|---|---|---|
| SMAPE | 11.907 | 11.920 | 11.927 | 11.926 | 11.925 | 11.925 | 11.950 | 11.890 | 11.728 | 11.910 | 11.901 | 0.059 |
| MASE | 1.612 | 1.610 | 1.653 | 1.603 | 1.619 | 1.620 | 1.625 | 1.623 | 1.613 | 1.591 | 1.617 | 0.016 |
| OWA | 0.870 | 0.872 | 0.872 | 0.872 | 0.872 | 0.872 | 0.849 | 0.862 | 0.876 | 0.870 | 0.868, | 0.009 |

Table S16: Short-term Forecasting Task on M4. The results are reported with different types of text prototypes.

Considering why the type of text prototype does not significantly affect results, we figure that in high dimensional space, almost all vectors are pairwise orthogonal Hopcroft & Kannan (2013). Which means that, in high-dimensional space, it is easy to generate a large number of almost orthogonal

vectors to represent different attributes. Thus, randomly selecting the same number of vectors, the represented space size and expressed number of features are almost the same. Therefore, the key is the number rather than the type.

In terms of probability, "two vectors orthogonal" is equivalent to "two vectors perpendicular" is equivalent to "two vectors uncorrelated" is equivalent to "$\cos\theta = 0$". For a $n$-dimensional space, randomly two vectors have: $\forall\epsilon, \lim_{n\to\infty} P(|\cos\theta| > \epsilon) = 0$. As shown in Figure S3, as the dimension increases, the probability of two random vectors being similar decreases. For LLM, $n > 1024, P(\theta = 0) < 0.00001$.

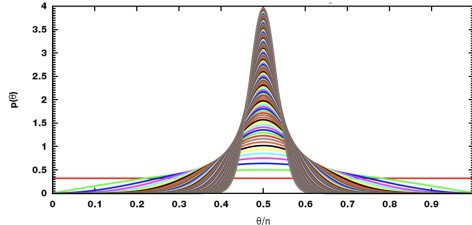

Figure S3: Probability Density of the Angle between Two Random Vectors in n-dimensional Space

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
