# OpenReview forum: "TEST: Text Prototype Aligned Embedding to Activate LLM's Ability for Time Series"
_ICLR.cc/2024/Conference — ICLR 2024 poster_

### Official Review · Reviewer_t5Ny · 2023-10-30

**Soundness:** 2 fair
**Presentation:** 2 fair
**Contribution:** 3 good
**Rating:** 6
**Confidence:** 3

**Summary:**

The authors propose an interesting way to do time series forecasting and classification with pretrained Large Language Models (LLMs). Whereas LLMs are designed to be used via prompts, the authors propose to find time series embeddings that are introduced in the model instead of finding suitable prompts. In this way, by considering an end-to-end approach, where the embeddings are adjusted according to the given dataset keeping the LLM weights frozen, the authors show that the proposed methodology does present a competitive performance. In particular, the larger the model, i.e. LLaMa-13B, the better the performance, potentially outperforming SOTA models like transformers. Finally, the authors present a brief analysis on the embeddings learn by the model, which suggests that these are mainly composed by adjectives/sentiments.

**Strengths:**

The authors motivate the proposed approach by clarifying that there are basically two ways to extend LLMs to time series: either by 1/creating a LM from a large enough time series corpus, which would imply an non-negligible amount of computational resources, or 2/ adjusting time series so that one can leverage available LLMs. Given that the later approach is more accessible to practitioners, the authors then start providing a way to embed time series to leverage existing LLMs for time series.

The strongest point of the paper is the notion of introducing time series embeddings to LLMs rather than finding the right prompt for it. Interestingly, the authors consider that by freezeing the weights of the LLM one can obtain embeddings that are leading to a competitive performance in terms of forecasting and classification.

Perhaps the most interesting point would be to understand the kind of embeddings that are obtained from the proposed methodology.

**Weaknesses:**

1. Given that the proposed approach relies on contrastive learning, I would suggest the authors to provide a more self-contained and better written text when it comes to contrastive learning. Currently the paper assumes a non-trivial amount of familiarity to the topic and makes it difficult to follow. For instance, what does the following term mean: _jitter-and scale strategy_?

2. There is no code included. This is relevant as I wonder if one can reproduce the experiments given the current writing quality of the paper.

3. There are no evaluations for zero-shot evaluations. According to the motivation of the paper, it might be relevant as well to consider the case where the time series embedding is developed in a particular dataset, and then tested in another one. Although this might not be necessarily too far from the few-shot evaluation already presented.

4. There is no analysis on time execution. The authors do not report how long it takes to obtain the time series embeddings.

5. The results are presented in an aggregated way across datasets. It would be of interest for researchers and practitioners how the proposed approach performs per dataset.

**Questions:**

1. What is jitter-andscale strategy?

2. can the authors please expand a bit on what the following line means? _In this work, we choose P representative text embedding tp as pivots/prototypes, and map TS embedding to them_

3. typo: Figure 3.g: spesific → specific

4. typo: Figure 2: Fine-turn → Fine-tune.

5. it seems that a general trend is that the LLaMa-13B performs best. It I understand correctly, this is as well the largest pretrained model considered. Further, one can see that for GPT2, the larger the model, the better the performance. Do we know, or can we do a conjecture, on what is the limiting performance in terms of model size (number of parameters)?

6. Why is LLM-QA not showing any variance? Or why is it depicted in this way?

7. Why is one-fits-all not used in *all* forecasting experiments? If I understand correctly, one can use it for comparison.

8. Why are not other models considered here? for instance DeepAR, MQCNN, Wavenet, NHiTS, NBeats, etc for these experiments? Even models like ARIMA, ETS, seasonal naive would be interesting to consider.

9. From 3.d it seems that the prompt length should be restricted to not be particularly large, but suprisingly the smaller the number of parameters (GPT-117M) the better the performance. How does this fit with the previous observation that larger models perform better?

10. On Embeddings (section 4.3): how does the embeddings change per model? For instance, what are the embeddings when using BERT-335M, GPT2-774M, etc?

11. When it comes to forecasting, is the proposed approach a point-wise or a probabilistic forecaster?

12. What is the performance if the LLM taken is the one of the first checkpoint, i.e. when the weights of the model are random? What is the gap between these case those taken for the current experiments?

13. What is the time execution/complexity of the proposed approach. Does this depend on the pretrained model in consideration? For instance, would it take longer of LLaMa-13B than for BERT-110M?

14. What is the performance per dataset when it comes to time series forecasting? The current experimental setting only reports results aggregated across different datasets.

---

> ### Author Response · Authors · 2023-11-15
> **Response to Reviewer t5Ny**
>
> We are extremely grateful for your review. You have raised many important issues. We agree with your comments and have modified our manuscript accordingly.
>
> 1.	Details about related contrastive learning.
>
> Response: We present more related work in Appendix 1.
>
> 2.	Jitter and scale strategy.
>
> Response: Sorry, given space limitations, some details have not been elaborated (Blue font in paragraph 1 of page 4). Here, the strategy is used for time series data augmentation: Weak augmentation is a jitter-and-scale strategy. We add random variations to the signal and scale up its magnitude; Strong augmentation is a permutation-and-jitter strategy, where permutation includes splitting the signal into a random number of segments with a maximum of M and randomly shuffling them. Please refer to: Eldele et al. Time-Series Representation Learning via Temporal and Contextual Contrasting. IJCAI 2021.
>
> 3.	The selection of prototypes.
>
> Response: The prototype is selected randomly. The prototypes (value, shape, frequency) shown in Figure 1 is an example. As in high dimensional space, almost all vectors are pairwise orthogonal, the number of prototypes rather than the type does matter (Blue font in paragraph 6 of page 5). Thus, by randomly selecting the same number of vectors, the represented space size and expressed number of features are almost the same (Appendix 2.5.2).
>
> Further, time series does not have text description. How to align objective data (time series, collected from complex systems) with subjective data (text, created and can be perceived by humans) is a topic worth studying but currently unresolved. This work aligns embedding spaces of time series and text through similarity, although there is no semantics and somewhat rigid, it is an enlightening attempt.
>
> 4.	Embeddings of each model.
>
> Response: The embeddings are learned from text-prototype-aligned contrastive learning. For different models, we randomly select P text token embedding vectors from the dictionary of LLM’s embedding layer as prototypes. As shown in Table 2, the embedding dimensions of different LLMs are different, e.g., 748, 1024.
>
> 5.	Forecasting mode.
>
> Response: The forecasting mode is the same as LLM, with tokens output sequentially, but our time series token is a subsequence, like subsequence-wise.
>
> 6.	Report all results.
>
> Response: You can check them in Appendix 2.
>
> 7.	Zero-shot evaluations.
>
> Response: Zero-shot forecasting experiments are in Appendix 2.2.5. Compared to TS transformers, our model has better results.
>
> 8.	The limiting performance in terms of model size.
>
> Response: Our experiment shows that when the size reaches about 300M, its accuracy is comparable to SOTA TS models (Blue font in paragraph 4 of page 7). And we agree with your insight. The boundaries of model size, data size, and their ratio are very interesting and valuable research topic. We are conducting research on it now.
>
> 9.	About LM-QA.
>
> Response: As you have considered, different prompts can lead to different results. In this work, we used the prompt templates by referring to Promptcast (Blue font in paragraph 3 of page 7).
>
> Xue et al. Promptcast: A new prompt-based learning paradigm for time series
> forecasting. 2023.
>
> 10.	Compare with one-fits-all.
>
> Response: In the original version, we compared with one-fits-all named GPT4TS as shown in Figure 3f (blue font in paragraph 3 of page 7). And when we implemented our method, one-fits-all only had a preprinted version and the code was not available. Thus, we directly used results they reported without rerunning code. In this revision, we added detailed results in Appendix.
>
> 11.	Compare with other models.
>
> Response: Thanks for your reminder. In this revision, we have added more compared models in Appendix. And we’d like to demonstrate that during our tests on the listed datasets, the methods you mentioned did not perform as well as the methods we compared. You can refer to: Wu et al. TimesNet: Temporal 2D-Variation Modeling for General Time Series Analysis. ICLR 2023.
>
> 12.	Time execution of encoder.
>
> Response: The training of encoders for time series embeddings only needs to use the text embedding obtained from LLM’s embedding layer and does not require training LLM. But LLMs are needed for training their head and inference. The encoder is a causal TCN (Blue font in paragraph 1 of page 7) and its training is fast, usually requiring only learn whole dataset once. Its detailed structure is in Appendix 2.1.1.
>
> 13.	The related code.
>
> Response: Related code and prompts will definitely be released.
>
> 14.	Figure 3.d.
>
> Response: Sorry, it’s a typo. The y axis should represent classification accuracy, we have corrected it.
>
> 15.	The performance if using LLM with random parameters.
>
> Response: This is an interesting thought, but it is not applicable to our approach as the random model does not have a trained embedding dictionary to use as prototypes.
>
> 16.	Typo.
>
> Response: Thanks for your careful checks. We have corrected typos.

---

> ### Author Response · Authors · 2023-11-22
> **Looking forward to further reply**
>
> Dear Reviewer t5Ny,
>
> We appreciate your previous thorough advice. You have raised a number of important issues. We have modified our manuscript accordingly and given a point-by-point response to your concerns and suggestions.
>
> To improve our work, we are very much looking forward to your further reply to our response. And we hope to have more time to make appropriate revisions based on your further suggestions.

---

> ### Comment · Reviewer_t5Ny · 2023-11-22
>
> I would like to thank the authors for their replies. I have increase my score by one point - my apologies for the previous answer. It was a mistake on my side.

---

> > ### Author Response · Authors · 2023-11-22
> > **Looking forward to more explanations and comments**
> >
> > Dear Reviewer t5Ny,
> >
> > We are very delighted to receive your reply! Thank you for your early comments as well.
> >
> > We believe we have answered all of your issues, including providing related work reviews, methods details and analysis, additional experiment and result details, and related code in the supplementary material.
> >
> > However, if you could let us know about any issues that are still open or offer additional suggestions to help us improve our paper, that would be greatly appreciated.
> >
> > Again, thank you for your time and patience. Very much looking forward for your next reply.

---

> > ### Author Response · Authors · 2023-11-23
> > **Thanks for your reply**
> >
> > We truly appreciate your encouragement, careful review, and valuable suggestions. Thanks again.

---

### Official Review · Reviewer_M1Ks · 2023-10-31

**Soundness:** 4 excellent
**Presentation:** 4 excellent
**Contribution:** 3 good
**Rating:** 8
**Confidence:** 2

**Summary:**

This research presents two approaches to handle Time-Series (TS) tasks with Large Language Models (LLMs): one that trains or fine-tunes a model for TS data, and another that transforms TS data for compatibility with existing LLMs. The focus is on the latter approach, introducing a method named TEST, which casts TimeSeries data into a format that LLMs can understand and then utilizes the LLM for TS tasks with the help of soft prompts. Experiments reveal that frozen LLMs equipped with the TEST strategy match or surpass current state-of-the-art TS models, enabling them to process TS data without losing language capabilities.

**Strengths:**

The paper made clear the difficulty of using LLM for TS data: "In this way, the core is to create embeddings that the LLM can understand."
TEST method proposed a "similarity-based, instance-wise, feature-wise, and text-prototype-aligned embedding for TS tokens" for this problem.

The explanation and writing about the method, as formally shown in Algorithm-1, is straight-forward clear.

The theretical analysis about casting Time-Series data into a pattern, and then translating the pattern into text, is an interesting hypothetis.

The experiment design, which includes 7 dimensions and 5 baseline models, is comprehensive and convincing. The improvements of TEST can be clearly visualized by the shaded pink area in top-left Figure-3, which is quite significant.

**Weaknesses:**

A picture to illustrate the connection between the "Time-series" data, and then the corresponding "text data to LLM", will be helpful for readers to understand the input and output of the TEST method. And it would be interesting to investigate the connection between resulting target text's semantics and the actual content of "Time-series" data.

**Questions:**

Do you have concrete examples of how a Time-Series data is transformed into text/(any format consumable by LLM), and the output of an LLM to such time-series data(of course, with the soft-prompt included)?

It would be helpful to include such information, if not in main text, in the Appendix.

How scalable is the proposed method.
Will you be releasing the encoder model, along with the soft-prompt for the task that the paper used?

---

> ### Author Response · Authors · 2023-11-15
> **Response to Reviewer M1Ks**
>
> We truly appreciate your encouragement, careful review, and valuable suggestions. We have given more detailed descriptions of related work and experiments in the Appendix. Below we give a point-by-point response to your concerns and suggestions.
>
> 1.	The connection between time series and text.
>
> Response: We agree with your comment. The ideal scenario is for time series subsequences to have corresponding text semantics. Figure 4 shows a case. However, as time series data does not have textual description, this work aligns the time series embedding space and the text embedding space through similarity but not semantics. It is somewhat rigid. Thus, selecting different text prototypes will cause a subsequence to correspond to different texts.
> Further, unlike images in multimodal alignment that can have text descriptions, time series data does not have corresponding text descriptions, how to align objective data (e.g. time series, collected from objective complex systems) with subjective data (e.g. text data, created by humans and can be perceived by humans) is a topic worth studying but currently unresolved.
>
> 2.	How scalable is the proposed method.
>
> Response: The related code and the trained prompts will definitely be released. In current work, encoders and prompts are both dataset-specific. That is, different time series datasets require training different encoder and soft prompt parameters.
> Different from the subjective text data, which can be perceived by humans, time series is more objective. The data from different scenarios varies greatly. Thus, it’s hard to build a fundamental model for all types of time series. This means models for medical time series data may be difficult to use in meteorological time series data. For example, RNN-based models are suitable for modeling medium and low-frequency data (e.g. vital signs, having sequence dependency), CNN-based models are suitable for modeling high-frequency data (e.g. electrocardiogram, having patterns), Transformer-based models are suitable for modeling long-term medium frequency data (e.g. electricity, weather, having periodic and trend characteristics). In this work, we are exploring the ability of existing LLM to solve time series problems. We do not recommend fine-tuning LLM to time series data as it will result in a loss of language ability. We recommend using additional adapters, such as an external encoder in this work, to enhance its time series analysis capabilities. Of course, we still believe that the current mainstream approach for implementing time series tasks is the traditional model, and the general fundamental model for is the crucial future work.

---

> > ### Comment · Reviewer_M1Ks · 2023-11-22
> >
> > The question is that how scalable is the prompts this paper uses, or the method to create the prompt, can be scaled towards other freezed LLMs to perform time-series tasks. The results in this paper is a bit of too good to be true, and as mentioned by another reviewer, being able to test one of the used prompts on some open-source model will really make a difference for the reception of this paper.

---

> > > ### Author Response · Authors · 2023-11-22
> > > **Response to Reviewer M1Ks**
> > >
> > > Thanks for your comments!
> > >
> > > The prompts matrices are soft, dataset-specific, and model-specific. They need to be retrained for each dataset and each used-llm, and have no semantics. They are initialized into a matrix consisting of randomly selected p (usually 10) vectors from the text embedding. e.g. 10*768 for gpt-2.
> > >
> > > The relevant code and appendix has been uploaded to the supplementary material.

---

### Official Review · Reviewer_YYDc · 2023-11-01

**Soundness:** 2 fair
**Presentation:** 3 good
**Contribution:** 3 good
**Rating:** 5
**Confidence:** 3

**Summary:**

The paper explores two ways to accomplish Time-Series (TS) tasks using Large Language Models (LLMs): LLM-for-TS (model-centric) and TS-for-LLM (data-centric). The authors focus on the TS-for-LLM approach and propose a method called TEST (TimE Series tokens to align the Text embedding space of LLM). TEST tokenizes TS data, embeds it using instance-wise, feature-wise, and text-prototype-aligned contrast, and aligns the TS embedding space with LLM's embedding layer space. Soft prompts are then created to make LLM more receptive to the embeddings. The paper demonstrates the feasibility of TS-for-LLM through theoretical analysis and experiments on TS classification, forecasting, few-shot, and representation tasks. The results show that pre-trained LLMs with the TEST strategy can achieve better or comparable performance to state-of-the-art TS models and offer benefits for few-shot learning and generalization. The authors hope this study will serve as a foundation for future research in the TS+LLM domain.

**Strengths:**

1. The organization of this paper is well-structured, making it easy to read and comprehend.
2. The paper proposes a novel embedding method (TEST) that aligns Time Series tokens with the text embedding space of Large Language Models, effectively enabling LLMs to handle Time Series data without compromising their language abilities. This study will serve as a foundation for future research in the TS+LLM domain.
3. Through experiments on various Time-Series tasks, the paper demonstrates that the TEST strategy can achieve better or comparable performance to state-of-the-art models, offering benefits in few-shot learning and generalization, thus validating the feasibility of the TS-for-LLM approach.

**Weaknesses:**

1. Based on the experience with LLM and multimodal fusion, it is possible to integrate LLM with image/video/speech representations using techniques such as adapters once good representations are obtained. Similarly, LLM4TS and TS4LLM can be fused with each other, for instance, when we obtain a good TS token embedding using TS4LLM, we can directly fine-tune the large model using LLM4TS, thereby achieving good performance. Therefore, is it appropriate to isolate LLM+TS into LLM4TS and TS4LLM as two distinct categories?
2. Time series prediction is a crucial task, and as observed from the experiments in this paper, there is a significant gap between TS4LLM, LLM4TS, and traditional methods when performing this task. Have the authors considered analyzing the reasons for this discrepancy?
3. In the statement "...we choose P representative text embedding tp as pivots/prototypes...", what specific type of text does P represent?

**Questions:**

see above

---

> ### Author Response · Authors · 2023-11-15
> **Response to Reviewer YYDc**
>
> We are extremely grateful for your review of the manuscript. You have raised a number of important issues. We agree with your comments and have modified our manuscript accordingly. We have given more detailed descriptions of related work and experiments in the Appendix. Below we give a point-by-point response to your concerns and suggestions.
>
> 1.	The categories of LLM4TS and TS4LLM.
>
> Response: We agree with the situation you mentioned. From a technical perspective, these two ideas can be integrated and may work better together. But we give these two categories to highlight different focuses: changing the model more or changing the data more. We want to present our views that fine-tuning LLM to TS data is unnecessary and not worth the loss. The method of changing data rather than models is more practical. The categories are more like purpose-wise instead of technology-wise.
> But your suggestions will greatly help to improve the academic rigor of our paper. In this revision, we have provided footnotes for categories (The footnote in blue on page 1). From a technical perspective, LLM+TS can be achieved by pre-training, fine-tuning, tool-augmented methods, external encoders, and their ensemble.
>
> 2.	LLM+TS v.s. traditional methods.
>
> Response: We have added more models, please refer to the Appendix for more detailed experiments.
>
> Different from the subjective text data, which can be perceived by humans, time series is more objective. The data from different scenarios varies greatly. Thus, it’s hard to build a fundamental model for all types of time series. This means models for medical time series data may be difficult to use in meteorological time series data. For example, RNN-based models are suitable for modeling medium and low-frequency data (e.g. vital signs, having sequence dependency), CNN-based models are suitable for modeling high-frequency data (e.g. electrocardiogram, having patterns), Transformer-based models are suitable for modeling long-term medium frequency data (e.g. electricity, weather, having periodic and trend characteristics). In this work, we are exploring the ability of existing LLM to solve time series problems. We do not recommend fine-tuning LLM to time series data as it will result in a loss of language ability. We recommend using additional adapters, such as an external encoder in this work, to enhance its time series analysis capabilities. Of course, we still believe that the current mainstream approach for implementing time series tasks is the traditional model, and the general fundamental model for is the crucial future work.
>
> 3.	The selection of prototypes.
>
> Response: The prototype is selected randomly. The prototypes (value, shape, frequency) shown in the figure is an example. As in high dimensional space, almost all vectors are pairwise orthogonal, the number of prototypes rather than the type does matter (Blue font in paragraph 6 of page 5). Thus, by randomly selecting the same number of vectors, the represented space size and expressed number of features are almost the same (Appendix 2.5.2).
>
> Further, time series data does not have textual description. How to align objective data (e.g. time series, collected from objective complex systems) with subjective data (e.g. text data, created by humans and can be perceived by humans) is a topic worth studying but currently unresolved. This work aligns the time series embedding space and the text embedding space through similarity, although there is no semantic relationship and it is somewhat rigid, it is also an enlightening attempt.

---

> ### Author Response · Authors · 2023-11-22
> **Looking forward to further reply**
>
> Dear Reviewer YYDc,
>
> We appreciate your previous thorough advice. You have raised a number of important issues. We have modified our manuscript accordingly and given a point-by-point response to your concerns and suggestions.
>
> To improve our work, we are very much looking forward to your further reply to our response. And we hope to have more time to make appropriate revisions based on your further suggestions.

---

### Official Review · Reviewer_3p6a · 2023-11-02

**Soundness:** 2 fair
**Presentation:** 2 fair
**Contribution:** 3 good
**Rating:** 5
**Confidence:** 3

**Summary:**

The paper introduces TEST, a TS-for-LLM method that leverages LLMs in time series (TS) data analysis without the need for fine-tuning. TEST focuses on transforming TS data into LLM-understandable embeddings. It shows potential in enhancing few-shot learning performance by harnessing the inherent knowledge within LLMs. The method involves training an encoder using contrastive learning, comprising three objectives: instance-wise, feature-wise, and text-prototype objectives. The instance-wise objective pulls closer to similar instance embeddings. The feature-wise objective is applied to each feature column to further discriminate instances. The text-prototype objective aims to align the TS embeddings and prototype embedding using feature-wise loss. Learnable soft prompts are also employed to assist LLMs in understanding time series data. Experimental results demonstrate that TEST can rival or outperform current SOTA methods.

**Strengths:**

1. The exploration of using language models for time series data without extensive pre-training or fine-tuning is an intriguing avenue of research.
2. I am impressed by the non-fine-tuning approach to achieve performance comparable to a supervised baseline. It demonstrates the effectiveness of the proposed method on the time-series problem. Additionally, exploring the TS-for-LLM method is a promising direction for few-shot settings.

**Weaknesses:**

1. The paper lacks clarity regarding how the representative prototypes are chosen. There are three groups of prototypes (value, shape, frequency), and it's unclear how these are determined. Does their selection depend on the dataset, and what criteria are used to assess their effectiveness?
2. I would like to see an examination of the performance impact of removing the decoder component. It looks like the decoder is not a necessary component in the proposed method.
3. The paper could be stronger from an ablation study to thoroughly investigate the effectiveness of the proposed components and objectives. For example, what's the performance without using soft-prompt or feature-wise objectives? This would help readers assess the contributions of each component.

**Questions:**

Have you considered the potential for parameter-efficient fine-tuning methods, such as LoRA, to outperform the use of soft prompts? Similar to using soft-prompts, LoRA does not require updating the model's entire parameter set.

Reference:
LoRA: Low-Rank Adaptation of Large Language Models, Hu et. al.

---

> ### Author Response · Authors · 2023-11-15
> **Response to Reviewer 3p6a**
>
> We are extremely grateful for your review of the manuscript. You have raised a number of important issues. We agree with your comments and have modified our manuscript accordingly. We have given more detailed descriptions of related work and experiments in the Appendix. Below we give a point-by-point response to your concerns and suggestions.
>
> 1.	The selection of prototypes.
>
> Response: The prototype is selected randomly. The prototypes (value, shape, frequency) shown in the figure is an example. As in high dimensional space, almost all vectors are pairwise orthogonal, the number of prototypes rather than the type does matter (Blue font in paragraph 6 of page 5). Thus, by randomly selecting the same number of vectors, the represented space size and expressed number of features are almost the same (Appendix section 2.5.2).
>
> Further, time series data does not have textual description. How to align objective data (e.g. time series, collected from objective complex systems) with subjective data (e.g. text data, created by humans and can be perceived by humans) is a topic worth studying but currently unresolved. This work aligns the time series embedding space and the text embedding space through similarity, although there is no semantic relationship and it is somewhat rigid, it is also an enlightening attempt.
>
> 2.	The impact of the encoder.
>
> Response: The decoder is the head of LLM. We use it as a pre-training trick. Of course, it is also possible to not train it during contrastive learning.
>
> 3.	Ablation study.
>
> Response: We agree with your comment. In this revision, we added more ablation experiments in Appendix section 2.5.
>
> 4.	The use of PEFT methods.
>
> Response: We agree with your comment. However, in our previous experiments, LoRA was not as good as p-tuning in the embedding layer. Personally, I think LoRA is more suitable for self-attention blocks.
>
> And your suggestion is also what we are considering. We suggest combining LLM and TS in a more effective way, rather than directly fine-tuning. In this way, it’s more possible to enable LLM to have both language and TS analysis capabilities. Thus, beyond this work, we are exploring a plugin-based paradigm of LLM+TS. Plugins are likely to be the additional adapters or external encoders, implemented by using the PEFT methods, like LoRA. The current experiment results show that the PEFT-based plugins can also produce nice results.

---

> ### Author Response · Authors · 2023-11-22
> **Looking forward to further reply**
>
> Dear Reviewer 3p6a,
>
> We appreciate your previous thorough advice. You have raised a number of important issues. We have modified our manuscript accordingly and given a point-by-point response to your concerns and suggestions.
>
> To improve our work, we are very much looking forward to your further reply to our response. And we hope to have more time to make appropriate revisions based on your further suggestions.

---

### Meta-Review · Area_Chair_6iNP · 2023-12-12

**Metareview:**

The paper proposes TEST, a novel method for aligning time series data with the text embedding space of pre-trained Large Language Models (LLMs) without the need for fine-tuning, enabling LLMs to effectively undertake time series tasks. Employing contrastive learning and soft prompts, the authors argue that TEST can match or surpass SOTA models for time series analysis while preserving the LLMs' linguistic capabilities.

Strengths highlighted across reviews include the innovative approach to leveraging LLMs for time series data, the comprehensive experimental design, and the potential for significant impact on the field of time series analysis with LLMs. Reviewers commend the attention to the challenge of creating embeddings understandable by LLMs, the theoretical analysis provided, and the visualization of improvements over baseline methods.

On the other hand, there are concerns regarding the clarity and completeness of the methodological description, such as the specifics of the contrastive learning strategies and the prototype selection process. The paper's presentation would benefit from illustrations to aid comprehension, a more accessible exposition of technical details, and additional information on the embeddings' nature.

The authors have constructively engaged with reviewer feedback and expanded on their manuscript, providing further elaborations in the Appendix and addressing queries related to method particularities, scalability, and performance nuances per dataset. The supplementary material now includes related code and the prompts utilized, bolstering the paper's reproducibility.

Given the revisions and the authors' thorough responses, which align with the feedback provided, the submission offers a valuable contribution to the domain, and the work stands on solid ground theoretically and experimentally. Reviewers express overall positive sentiments, with concerns satisfactorily addressed by the authors through clarifications and additional experiments. Therefore, the manuscript is recommended for acceptance, with the expectation that the authors will incorporate the comments in their final version.

**Justification For Why Not Higher Score:**

There are concerns regarding the clarity and completeness of the methodological description, such as the specifics of the contrastive learning strategies and the prototype selection process. The paper's presentation would benefit from illustrations to aid comprehension, a more accessible exposition of technical details, and additional information on the embeddings' nature.

**Justification For Why Not Lower Score:**

Strengths highlighted across reviews include the innovative approach to leveraging LLMs for time series data, the comprehensive experimental design, and the potential for significant impact on the field of time series analysis with LLMs. Reviewers commend the attention to the challenge of creating embeddings understandable by LLMs, the theoretical analysis provided, and the visualization of improvements over baseline methods.

---

### Decision · Program_Chairs · 2024-01-16

Accept (poster)